# Thioredoxin-interacting protein regulates haematopoietic stem cell ageing and rejuvenation by inhibiting p38 kinase activity

Haiyoung Jung[1,2,*], Dong Oh Kim[1,2,*], Jae-Eun Byun[1,3], Won Sam Kim[1,2], Mi Jeong Kim[1,2], Hae Young Song[1], Young Kwan Kim[4], Du-Kyeong Kang[5], Young-Jun Park[1,2], Tae-Don Kim[1,2], Suk Ran Yoon[1,2], Hee Gu Lee[1,6], Eun-Ji Choi[7], Sang-Hyun Min[8] & Inpyo Choi[1,2]

Ageing is a natural process in living organisms throughout their lifetime, and most elderly people suffer from ageing-associated diseases. One suggested way to tackle such diseases is to rejuvenate stem cells, which also undergo ageing. Here we report that the thioredoxin-interacting protein (TXNIP)-p38 mitogen-activated protein kinase (p38) axis regulates the ageing of haematopoietic stem cells (HSCs), by causing a higher frequency of long-term HSCs, lineage skewing, a decrease in engraftment, an increase in reactive oxygen species and loss of Cdc42 polarity. TXNIP inhibits p38 activity via direct interaction in HSCs. Furthermore, cell-penetrating peptide (CPP)-conjugated peptide derived from the TXNIP-p38 interaction motif inhibits p38 activity via this docking interaction. This peptide dramatically rejuvenates aged HSCs *in vitro* and *in vivo*. Our findings suggest that the TXNIP-p38 axis acts as a regulatory mechanism in HSC ageing and indicate the potent therapeutic potential of using CPP-conjugated peptide to rejuvenate aged HSCs.

[1] Immunotherapy Convergence Research Center, Korea Research Institute of Bioscience and Biotechnology (KRIBB), Yuseong-gu, Daejeon 34141, Republic of Korea. [2] Department of Functional Genomics, University of Science and Technology, Yuseong-gu, Daejeon 34113, Republic of Korea. [3] Department of Biochemistry, School of Life Sciences, Chungbuk National University, Cheongju 28644, Republic of Korea. [4] Scripps Korea Antibody Institute, 1 Kangwondaehak-gil, Chuncheon 24341, Republic of Korea. [5] Bioenergy and Biochemical Research Center, Korea Research Institute of Bioscience and Biotechnology, Daejeon 34141, Republic of Korea. [6] Department of Biomolecular Science, University of Science and Technology, Yuseong-gu, Daejeon 34113, Republic of Korea. [7] Department of Hematology, Asan Medical Center, University of Ulsan College of Medicine, Seoul 05505, Republic of Korea. [8] New Drug Development Center, Daegu-Gyeongbuk Medical Innovation Foundation (DGMIF), 80 Chumbokro Dong-gu 41061, Daegu, Republic of Korea. * These authors contributed equally to this work. Correspondence and requests for materials should be addressed to I.C. (email: ipchoi@kribb.re.kr) or to H.J. (email: haiyoung@kribb.re.kr).

The ageing of stem cells underlies the ageing of tissues and represents a progressive decline in functional activities that maintain the homoeostasis and regeneration of the specific tissues in which stem cells reside[1,2]. In the haematopoietic system, haematopoietic stem cells (HSCs) continuously replenish blood cells, exhibiting a high turnover rate throughout life. The ageing of HSCs is likely the key process of decline in immune function with age or ageing-associated diseases and is driven by both extrinsic and intrinsic factors[3–5]. A series of studies have reported that aged mice exhibit remarkable changes in their haematopoietic systems, such as the expansion of $CD34^- Flk2^- LSK$ (lineage$^-$ c-kit$^+$ Sca-1$^+$) cells (LT-HSCs), lineage skewing, an increase in reactive oxygen species (ROS) and a decreased number of leucocytes in the peripheral blood (PB)[4–7]. HSC research groups have proposed several factors involved in HSC ageing, including mitochondrial DNA damage[8], ROS and p38 (refs 9,10), DNA damage[3], telomere shortening[11], epigenetic alteration[12], loss of Cdc42 polarity[4,13], Wnt5a[13], replication stress[14] and others[1]. Recent reports have also suggested possible mechanisms to rejuvenate aged HSCs via the reduction of Cdc42 activity using its inhibitor[4], SIRT3 overexpression[15] and prolonged fasting[16].

TXNIP is a known inhibitor of thioredoxin and is a tumour suppressor that blocks cell-cycle progression[17,18]. In our previous results, *TXNIP* was highly expressed in HSCs and its expression decreased as HSCs differentiated into lineage cells. *TXNIP* deficiency exhibited higher levels of ROS in HSCs and decreased HSC repopulation capacity. TXNIP acted as an antioxidant protein under oxidative stress by regulating p53 activity via direct interaction[19–21].

p38 is a Ser/Thr kinase that regulates the growth, proliferation, death and differentiation of cells in response to multiple stimuli[22,23]. Many researchers have observed p38 activation in various pathological conditions or during cellular ageing via elevated ROS, resulting in HSC defects. These researchers have also suggested that the pharmacological inhibition of p38 activity might restore the defects of HSCs *in vitro* and *in vivo*. For example, administration of SB203580, a p38 inhibitor, restored repopulation capacity, maintained the quiescence of HSCs and promoted the expansion of mouse or human HSCs *ex vivo*[1,9,11,22,24,25].

On the basis of our previous data, we inferred the regulatory function of TXNIP in HSC ageing[20,21]. In this study, we show that the loss of *TXNIP* induces the premature ageing of HSCs by elevating ROS production and inducing ageing-associated genes via upregulating p38 activity. We also show that TXNIP interacts with p38 via docking interaction and inhibits p38 activity in HSCs. Furthermore, we examine the potential of TXNIP-derived peptide to inhibit p38 activity to rejuvenate aged HSCs *in vitro* and *in vivo*. Altogether, we propose the novel functions of TXNIP in HSC ageing via regulating p38 activity and the possibility of the rejuvenation of aged HSCs via inhibiting p38 activity with TXNIP-derived peptide.

## Results

**Premature ageing of $TXNIP^{-/-}$ HSCs.** To identify the function of TXNIP in HSCs, we confirmed the expression of *TXNIP* in various subpopulations of mouse bone marrow (BM) cells. In agreement with our previous data[20,21], mRNA level of *TXNIP* was increased in LT-HSCs (Supplementary Fig. 1a). Next, to determine the effect of TXNIP on HSC ageing, we analysed white blood cells (WBCs) in the PB of $TXNIP^{+/+}$ and $TXNIP^{-/-}$ mice at 2 (young), 6, 12 and 24 (old) months of age[3,6,26]. $TXNIP^{-/-}$ mice showed dramatically skewed differentiation to myeloid at the age of 12 months, even more than old $TXNIP^{+/+}$ mice (Supplementary Fig. 1b). Ageing-associated phenotypes of

12-month-old $TXNIP^{-/-}$ mice in haematopoiesis were also observed in their LT-HSC frequency of LSKs in BM cells, which were comparable to those of old $TXNIP^{+/+}$ mice (Fig. 1a,b). Next, we analysed the frequency of BM cells and absolute number of LT-HSCs, ST-HSCs and MPPs in each mouse. $TXNIP^{-/-}$ mice showed the exhaustion of HSCs at 12- and 22-month age (Supplementary Fig. 1c,d). Additionally, we investigated the ratio of WBCs in the PB of 12-month-old $TXNIP^{+/+}$ and $TXNIP^{-/-}$ female mice to confirm the ageing phenotype of female mice. Twelve-month-old $TXNIP^{-/-}$ female mice also showed markedly skewed differentiation to myeloid (Supplementary Fig. 1e). These data strongly implied the possibility of HSC ageing in 12-month-old $TXNIP^{-/-}$ mice, similar to that in old $TXNIP^{+/+}$ mice. Next, to obtain direct evidence of HSC ageing in 12-month-old $TXNIP^{-/-}$ mice, we investigated levels of ROS, which are elevated with age, and the expression of ageing-associated genes in HSCs. We selected four representative genes from previous reports, including *p16*, *p19*, *p21* and *Wnt5a*, which play critical roles in HSC ageing[1,13,27]. As expected, 12-month-old $TXNIP^{-/-}$ HSCs exhibited higher ROS (Fig. 1c) and an induction of *p16*, *p19*, *p21* and *Wnt5a* comparable to that of old $TXNIP^{+/+}$ HSCs (Fig. 1d–g).

To determine the roles of TXNIP on haematopoiesis, we administered 5-fluorouracil (5-FU), which targets cycling cells, leading to a transient leucopenia in the blood. In addition, we administered NAC (N-acetyl-L-cysteine), an antioxidant agent, to examine whether the increased ROS production contributes to the higher sensitivity of $TXNIP^{-/-}$ HSCs to 5-FU treatment[28,29]. $TXNIP^{+/+}$ mice recovered to normal status after 14 days in WBC counts (Supplementary Fig. 1f), but all $TXNIP^{-/-}$ mice gradually died. Interestingly, NAC treatment fully rescued $TXNIP^{-/-}$ mice from 5-FU-induced leucopenia (Supplementary Fig. 1g). 5-FU treatment highly induced the production of ROS in $TXNIP^{-/-}$ HSCs than $TXNIP^{+/+}$ HSCs and the induction of ROS was reversed by NAC treatment (Supplementary Fig. 1h). These results suggest that the increased production of ROS in $TXNIP^{-/-}$ HSCs may result in the defects in the repopulation capacity of HSCs under haematopoietic stress.

Next, we performed a competitive transplantation assay, which is a standard assay for determining autonomous HSC function[4]. We isolated $CD45.2^+$ LT-HSCs from young $TXNIP^{+/+}$ and $TXNIP^{-/-}$ mice and transplanted with competitor BM cells ($CD45.1^+$) into lethally irradiated congenic recipients ($CD45.1^+$). HSC engraftment was markedly decreased (Fig. 1h), and ageing-associated skewing to myeloid was observed in recipient mice receiving $TXNIP^{-/-}$ HSCs after 4 months (Fig. 1i). Analysis of BM cells also revealed a decrease in engraftment (Supplementary Fig. 1i) and higher LT-HSC frequency of LSKs (Fig. 1j and Supplementary Fig. 1j). As shown in our previous data[20], ROS levels were also higher in $TXNIP^{-/-}$ HSCs (Fig. 1k).

To determine the numbers of functional HSCs, we also performed a limiting dilution analysis of 12-month-old $TXNIP^{+/+}$ and $TXNIP^{-/-}$ HSCs and calculated the competitive repopulating units[30]. As expected, 12-month-old $TXNIP^{-/-}$ HSCs showed markedly reduced numbers of functional HSCs (Fig. 1l).

These data indicate that TXNIP may play a role in HSC ageing and that the loss of *TXNIP* may induce the premature ageing of HSCs by elevating ROS production and inducing ageing-associated genes.

**The activation of p38 in $TXNIP^{-/-}$ HSCs.** Our data and previous reports suggest that HSCs are exposed to oxidative stress with age via elevated ROS levels. One potential mediator of

oxidative stress is p38. As previously noted, p38 plays crucial roles in regulating stress-induced signalling cascades and ageing-associated gene expression in HSCs[9,31]. To identify the role of ROS on p38 activation in $TXNIP^{-/-}$ HSCs, we administered NAC to 12-month-old $TXNIP^{-/-}$ mice. ROS level and p38 activation were decreased in HSCs by NAC treatment

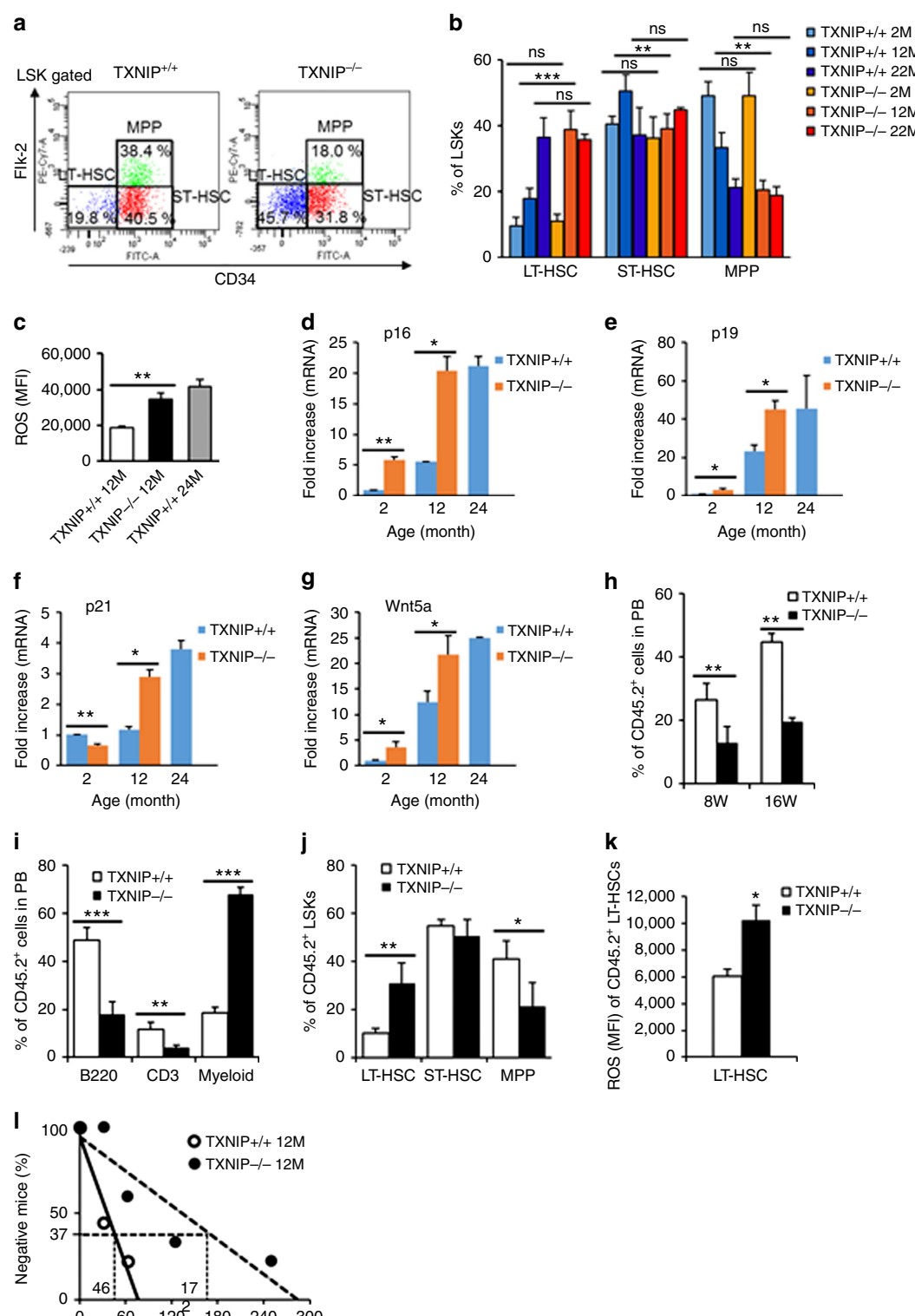

**Figure 1 | Premature ageing of *TXNIP* deficient HSCs.** (**a**) A representative image of LT-HSCs, ST-HSCs and MPPs among LSKs in 12-month-old mice. (**b**) LT-HSCs, ST-HSCs and MPPs among LSKs ($n = 6$). (**c**) ROS levels in LT-HSCs ($n = 5$). (**d–g**) Quantitative real-time PCR of *p16*, *p19*, *p21* and *Wnt5a* in LT-HSCs ($n = 3$). (**h,i**) Distribution of donor-derived WBCs in PB (**h**) and frequency of B220$^+$, CD3$^+$ and myeloid cells among donor-derived WBCs in PB (**i**) ($n = 5$ from two experiments). (**j,k**) Frequency of LT-HSCs, ST-HSCs and MPPs among donor-derived LSKs (**j**) and ROS levels of donor-derived LT-HSCs (**k**) ($n = 5$ from two experiments). (**l**) Limiting dilution analysis of HSCs from 12-month-old $TXNIP^{+/+}$ and $TXNIP^{-/-}$ mice ($n = 8$–10). Data are mean ± s.d. Statistical significance was determined using a two-tailed Student's $t$-tests. *$P < 0.05$, **$P < 0.01$, ***$P < 0.001$.

(Supplementary Fig. 2a,b). To understand the relationship between TXNIP and p38 in HSCs, we first confirmed the expression of *p38* isoforms (α, β, γ and δ)[22,23]. *p38α* was predominantly expressed and increased in LT-HSCs (Supplementary Fig. 2c).

Next, to confirm the relationship between TXNIP and p38 in HSCs ageing, we examined the levels of TXNIP and p38 activity in HSCs with age. TXNIP and p38 activity were increased in lin⁻ cells and HSCs with age and a loss of *TXNIP* resulted in p38 activation in HSCs (Fig. 2a,b and Supplementary Fig. 2d–f).

**TXNIP interacts with p38 directly in HSCs.** Our results imply that the interaction between TXNIP and the p38 pathway may regulate p38 activity in HSCs. We first tested the direct interaction between TXNIP and p38 using immunoprecipitation and an *in situ* proximity ligation assay (PLA). These two proteins directly interacted in BM cells and HSCs (Fig. 2c,d).

Next, to investigate the effect of ROS on their interaction, we administered $H_2O_2$. TXNIP was quickly induced and then

decreased, but p38 activity increased continuously up to 60 min in BM cells (Supplementary Fig. 2g). The interaction between TXNIP and p38 was increased by $H_2O_2$ treatment and ageing in HSCs (Fig. 2d). Glutathione S-transferase (GST) pull-down assay confirmed these results in TXNIP- and p38-overexpressed 293T cells (Supplementary Fig. 2h).

To examine the importance of p38 kinase activity on their interaction, we constructed a kinase-dead dominant-negative mutant for *p38α* (*p38AF*) by replacing the Thr-Gly-Tyr motif (activating phosphorylation sites) with Ala-Gly-Phe[32]. p38AF and treatment with SB203580 did not affect p38 interaction ability with TXNIP (Fig. 2e and Supplementary Fig. 2i). Previous studies have proposed that p38 complexed with its activator or substrate via a docking domain and recognized short docking motifs on the interaction partners. The docking motif contains basic residues and a hydrophobic-X-hydrophobic sub-motif $(K/R_{2-3}-X_{1-6}-\phi-X-\phi)$[33,34]. To determine the residues necessary for TXNIP-p38 interaction, we mutated four potential docking motifs in *TXNIP* by site-directed mutagenesis of hydrophobic

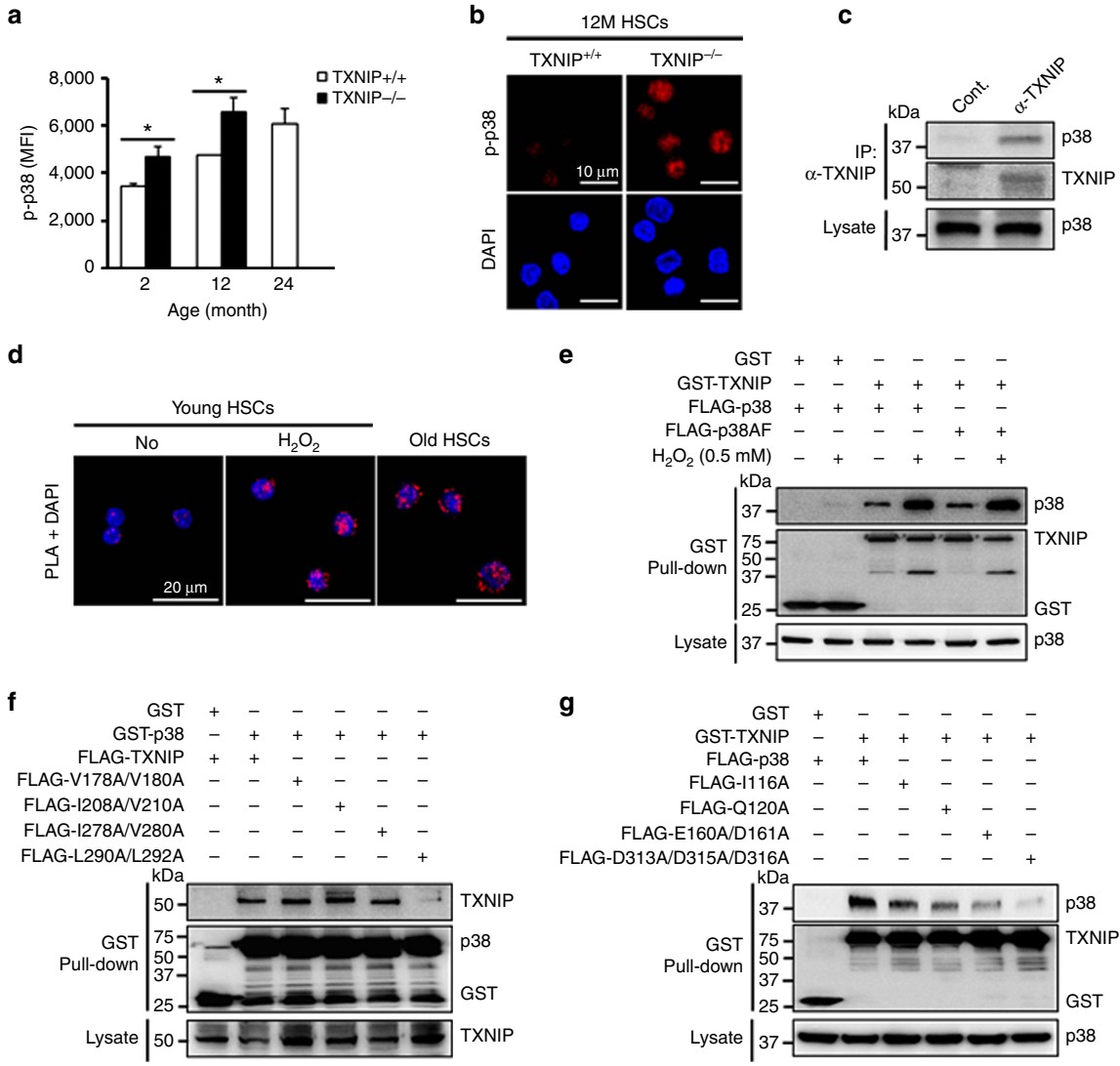

**Figure 2 | The activation of p38 in *TXNIP*⁻/⁻ HSCs and physical interaction between TXNIP and p38 in HSCs.** (**a**) Levels of p38 phosphorylation were determined by mean of fluorescence intensity using flow cytometry in LT-HSCs ($n = 3$ from two experiments). (**b**) Confocal images of phospho-p38 in 12-month-old LT-HSCs ($n = 3$ from two experiments). (**c**) Immunoprecipitation assay in young BM cells (repeated two times). (**d**) *In situ* PLA images in young LT-HSCs or old HSCs. Freshly isolated LT-HSCs were treated with 0.5 mM $H_2O_2$ for 1 h in HSC media (repeated two times). (**e**) GST pull-down assay in 293T cells (repeated two times). (**f,g**) GST pull-down assays in 293T cells (repeated three times). Data are mean ± s.d. Statistical significance was determined using a two-tailed Student's *t*-tests. *$P < 0.05$.

residues in sub-motifs. L290 and L292 residues of TXNIP were important for their interaction (Fig. 2f). p38 docking domain mutants decreased the interaction between TXNIP and p38 (Supplementary Fig. 2j and Fig. 2g)[34]. Furthermore, to confirm the unique interaction between TXNIP and p38 via a docking site, we mutated four residues of the TXNIP docking site, including basic residues, and mutated the Q120 (glutamine) of p38 to A (alanine). Their interaction was reduced markedly by these mutations (Supplementary Fig. 2k). These data demonstrate that TXNIP interacts with p38 directly via docking interaction and that its interaction may inhibit p38 activity in HSCs.

**Rejuvenation of HSCs by p38 inhibition *in vivo*.** To understand the roles of p38 in $TXNIP^{-/-}$ HSCs *in vivo*, we crossed $TXNIP^{-/-}$ mice with $p38^{AF/+}$ mice, which contained a dominant-negative allele and homozygous $p38^{AF/AF}$ embryos that died on approximately day 11.5, to generate $TXNIP^{-/-}/p38^{AF/+}$ mice[32,35]. To investigate whether p38 inhibition is critical to rejuvenate the defects of aged HSCs, we isolated $CD45.2^{+}$ LT-HSCs from indicated mice and then transplanted with competitor BM cells ($CD45.1^{+}$) into lethally irradiated congenic recipients ($CD45.1^{+}$). Surprisingly, the engraftment and skewing of WBCs were dramatically restored to a level comparable to those of young $TXNIP^{+/+}$ HSCs in $TXNIP^{-/-}/p38^{AF/+}$ HSCs (Fig. 3a,b). Also, a higher frequency of LT-HSCs returned to $TXNIP^{+/+}$ HSC levels in $TXNIP^{-/-}/p38^{AF/+}$ HSCs (Fig. 3c). SB203580 administration also had similar effects on $TXNIP^{-/-}$ and old HSCs. $TXNIP^{-/-}/p38^{AF/+}$ HSCs-received or long-term SB203580-administered recipients maintained low levels of p38 activity and ROS in their donor-derived HSCs (Fig. 3d,e). The skewing of WBCs in $TXNIP^{-/-}$ mice at the age of 12 months was also restored by crossing with the *p38* dominant-negative allele to a level comparable to that of young mice (Fig. 3f). Furthermore, $TXNIP^{-/-}/p38^{AF/+}$ mice survived with 5-FU treatment (Supplementary Fig. 3). Collectively, our data identified the function of p38 with respect to $TXNIP^{-/-}$ HSC ageing and indicated the possible application of p38 inhibitors as a rejuvenating drug for aged HSCs *in vivo*.

**TXNIP-derived peptide inhibits p38 activity.** As noted in a previous report, TXNIP recombinant was insoluble[36]. Therefore, we prepared a GST-fused truncation mutant for TXNIP (150.a.a–317.a.a) (GST-TXNIP-T), and p38 was prepared with a His-tagged protein. GST-TXNIP-T interacted with His-p38 and inhibited p38 kinase activity *in vitro* (Supplementary Fig. 4a,b). Next, to exclude pitfalls from the overexpression of fully cloned genes in cells, we designed a short peptide from the docking motif of TXNIP to target p38. We first generated green fluorescent protein (GFP) plus helix-forming peptide linkers, $(EAAAK)_5$, and *TXNIP*-derived peptide clones (Supplementary Fig. 4c). The lentiviral GFP-fused peptides were overexpressed in 293T cells, and the TN13 peptide with the highest affinity toward p38 was selected via immunoprecipitation assay (Supplementary Fig. 4d). To understand the complex between TN13 and p38, we designed a putative complex model from the peptide in the structure of p38 MAPK (PDB ID: 1LEW)[34]. In our putative model, TN13 interacted with the docking region of p38, and this interaction was very similar to those of other peptides (Supplementary Fig. 4e,f)[34]. Next, to investigate the physiological consequences of the interaction between TN13 and p38 in cells, we designed a fluorescein isothiocyanate (FITC)-linked peptide bearing a CPP. To deliver the synthesized peptide into cells, an HIV TAT transduction domain sequence (YGRKKRRQRRR) was linked to the N-terminus of the TN13 peptide, and to monitor the cell-penetrating efficiency of the peptide, FITC was linked to the N-terminus of the TAT sequence (TAT-TN13)[37,38]. The interaction between TAT-TN13 and His-p38 was determined by isothermal titration calorimetry (ITC), which measures the binding equilibrium directly by determining the heat evolved on association of a ligand with its binding protein[39], and their interaction was compared with TAT control (Supplementary Fig. 4g,h). To determine the specificity of TN13, we performed kinase assays for p38 isoforms *in vitro*. SB203580 completely inhibited kinase activity of two isoforms, p38α and p38β, but TN13 showed a specificity on p38α (Supplementary Fig. 4i). It suggests that TN13 may have high specificity on p38α isoform than SB203580 and this specificity may reduce the side effects *in vitro* and *in vivo*.

**TAT-TN13 rejuvenates aged HSCs *in vitro*.** We confirmed the cell-penetrating efficiency of TAT-TN13 into HSCs using confocal images (Supplementary Fig. 5a). Penetrated TAT-TN13 markedly inhibited p38 phosphorylation in old BM cells and old HSCs and was comparable to SB203580 (Fig. 4a,b). To identify the inhibitory mechanism of TAT-TN13, we investigated the interaction between p38 and its upstream kinases, MKK3 and MKK6, during TAT-TN13 treatment. TAT-TN13 treatment inhibited the interaction between p38 and MKK3 or MKK6 in 293T cells (Supplementary Fig. 5b,c). These results imply that TAT-TN13 shares the docking region of p38 for MKK3 or MKK6 and inhibits p38 activity via blocking the interaction between p38 and its upstream kinases. Our peptide also competed with TXNIP for interaction with p38 in BM cells and HSCs (Fig. 4c,d). These data also supported the docking interaction between TXNIP and p38. As shown in Fig. 2, our data suggested that the increased p38 activity with age was a critical change that resulted in HSC defects. We hypothesized that p38 inhibition in old HSCs via TAT-TN13 treatment may return HSCs to young states or at least improve their functional defects, as shown via SB203580 administration. To confirm the rejuvenating potential of TAT-TN13, we treated old HSCs with TAT-TN13 (10 µM) for 16 h *in vitro*. As expected, elevated p38 activity in old HSCs was dramatically reduced by TAT-TN13 treatment to levels comparable to those of SB203580 treatment, and ROS levels were decreased in TAT-TN13-treated old HSCs (Fig. 4e,f).

Recently, Geiger and his colleagues reported the role of Cdc42 activity in the cell-intrinsic ageing of HSCs. The authors proposed that the polarity of Cdc42 is a marker for differentiating young and old phenotypes of HSCs and that a pharmacological reduction of Cdc42 activity rejuvenates aged HSC function[4,13]. Therefore, we used the polarity of Cdc42 as an indicator of the rejuvenating phenotype of aged HSCs. After TAT-TN13 treatment for 16 h, HSCs were fixed and stained with Cdc42 antibody. Most depolarized old HSCs returned to polarized HSCs (Fig. 4g), and the expressions of ageing-associated genes were remarkably decreased in TAT-TN13-treated old HSCs (Fig. 4h–k). In addition, ageing-associated gene expression exhibited similar patterns after competitive transplantation assay *in vivo* (Supplementary Fig. 6a–d). Furthermore, TAT-TN13 treatment increased the homing ability of old HSCs after short-term transplantation (Fig. 4l). Taken together, our results indicate that the regulation of p38 activity might be a critical tool to rejuvenate aged HSCs, and we confirmed the potential of TAT-TN13 to inhibit p38 activity to rejuvenate aged HSCs *in vitro*.

**Rejuvenation of aged HSCs by GFP-TN13 and TAT-TN13 *in vivo*.** HSCs were isolated from mice and transduced three times with GFP-TN13-expressing lentiviral vector for 36 h. After incubation in HSC media, sorted $GFP^{+}$ HSCs were competitively transplanted into recipients (Fig. 5a). GFP-TN13 expression in

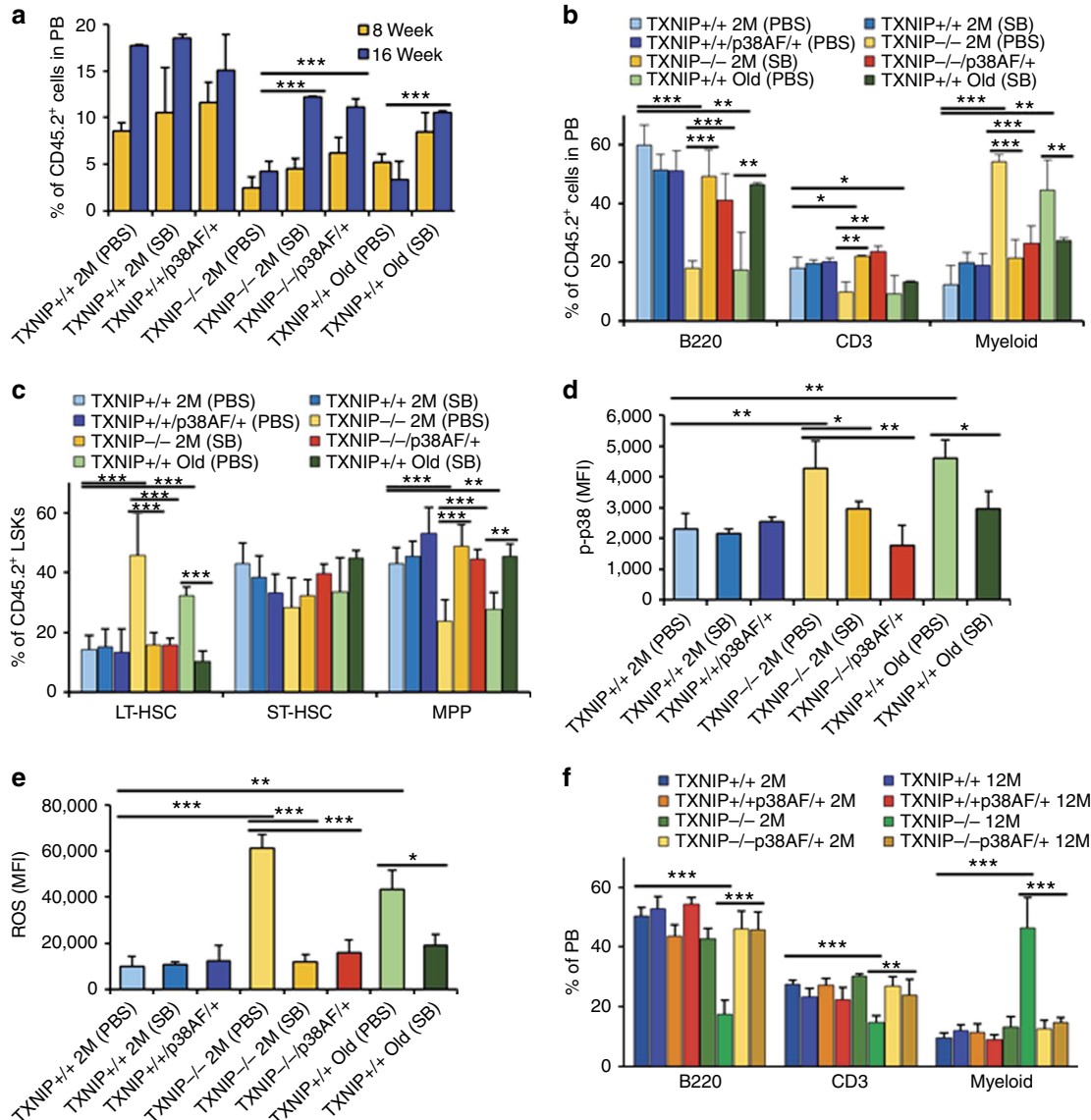

**Figure 3 | Inhibition of p38 activity rejuvenates the defects of *TXNIP*$^{-/-}$ and aged HSCs *in vivo*.** (**a**,**b**) Distribution of donor-derived WBCs in PB (**a**) and frequency of B220$^+$, CD3$^+$ and myeloid cells among donor-derived WBCs in PB (**b**) ($n = 6$–7 from two experiments). (**c**) LT-HSCs, ST-HSCs and MPPs among donor-derived LSKs ($n = 6$–7 from two experiments). (**d**) Levels of phospho-p38 in donor-derived LT-HSCs were determined using flow cytometry ($n = 6$–7 from two experiments). (**e**) ROS levels in donor-derived LT-HSCs ($n = 6$–7 from two experiments). (**f**) LT-HSCs, ST-HSCs and MPPs among LSKs in 2- or 12-month-old mice ($n = 5$–8). Data are mean ± s.d. Statistical significance was determined using a two-tailed Student's *t*-tests. *$P < 0.05$, **$P < 0.01$, ***$P < 0.001$.

12-month-old *TXNIP*$^{-/-}$ or old HSCs exhibited restoration of their ageing phenotypes (Fig. 5b–d). Elevated p38 activity in 12-month-old *TXNIP*$^{-/-}$ and old HSCs was dramatically reduced by GFP-TN13 expression to levels comparable to those of young HSCs, and ROS levels were decreased in GFP-TN13-transduced 12-month-old *TXNIP*$^{-/-}$ and old HSCs (Fig. 5e,f). Next, to investigate the possible application of TAT-TN13 as a rejuvenating drug for aged HSCs *in vivo*, we performed a competitive transplantation of TAT-TN13-treated old HSCs or 12-month-old *TXNIP*$^{-/-}$ HSCs. Both of them exhibited restoration of aged phenotypes of HSCs comparable to SB203580 treatment (Fig. 6a–c) and maintained low levels of p38 activity and ROS (Fig. 6d,e). To assess the effect of TAT-TN13 on haematopoietic stress in old HSCs, 5-FU (100 mg kg$^{-1}$) was i.p. injected into young and old mice, and on the next day, TAT-TN13 (25 mg kg$^{-1}$) was administered via i.p. injection daily

for 4 days. Surprisingly, TAT-TN13 administration restored the WBCs of old mice better than those of young mice at early time points and also saved their lives (Supplementary Fig. 7). Overall, these results demonstrated the potential of GFP-TN13 or TAT-TN13 as an applicable therapeutic tool for rejuvenating aged HSCs *in vivo* (Fig. 7).

## Discussion
Although stem cell ageing is complex and context dependent, stem cells apart from their environment share certain conserved molecular mechanisms in their ageing process. Recent studies have shown that impaired functions of aged stem cells were restored by pharmacological treatments, and these findings indicated new therapeutic targets for rejuvenating stem cells[1,4].

A potentially promising target molecule in stem cell ageing is p38, which is activated by ROS in aged stem cells. p38 activation

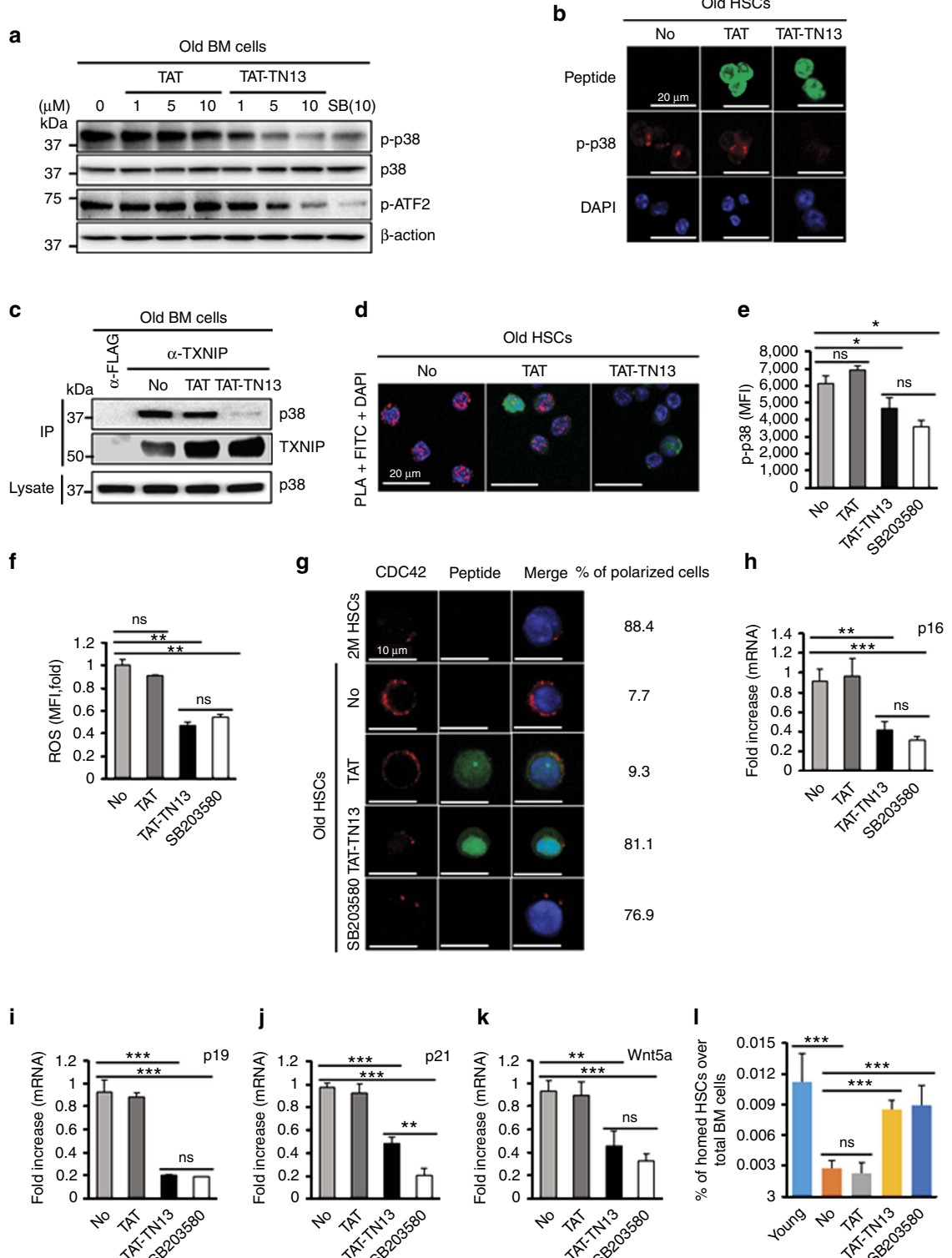

**Figure 4 | TXNIP-derived peptide (TAT-TN13) inhibits p38 activity and rejuvenates aged HSCs *in vitro*.** (**a**) Phosphorylation of p38 and ATF2 in old BM cells. Freshly isolated old BM cells were treated with TAT, TAT-TN13 or SB203580 for 1 h in RPMI 1640 containing 10% FBS (repeated two times). (**b**) Confocal images of phopho-p38 in old HSCs. Sorted old LT-HSCs were treated with 10 μM TAT or TAT-TN13 for 1 h in HSC media (repeated three times). (**c,d**) Immunoprecipitation assay in old BM cells (**c**) and *in situ* PLA images in old HSCs (**d**). Old BM cells and old LT-HSCs were treated with 10 μM TAT or TAT-TN13 for 1 h (repeated two or three times). (**e,f**) Levels of phospho-p38 (**e**) and ROS levels (**f**) in old HSCs. Sorted LT-HSCs were treated with 10 μM TAT, TAT-TN13 or SB203580 for 16 h in HSC media (*n* = 3 from two experiments). (**g**) Polar distribution of Cdc42 in LT-HSCs. Sorted LT-HSCs were treated with 10 μM TAT, TAT-TN13 or SB203580 for 16 h in HSC media (repeated two times). (**h–k**) Quantitative real-time PCR of ageing-associated genes in old HSCs. Sorted LT-HSCs were treated with 10 μM TAT, TAT-TN13 or SB203580 for 16 h in HSC media (*n* = 3 from two experiments). (**l**) The homing ability of old HSCs. Sorted LT-HSCs were treated with 10 μM TAT, TAT-TN13 or SB203580 for 16 h in HSC media (*n* = 6–10). Data are mean ± s.d. Statistical significance was determined using a two-tailed Student's *t*-tests. *$P < 0.05$, **$P < 0.01$, ***$P < 0.001$.

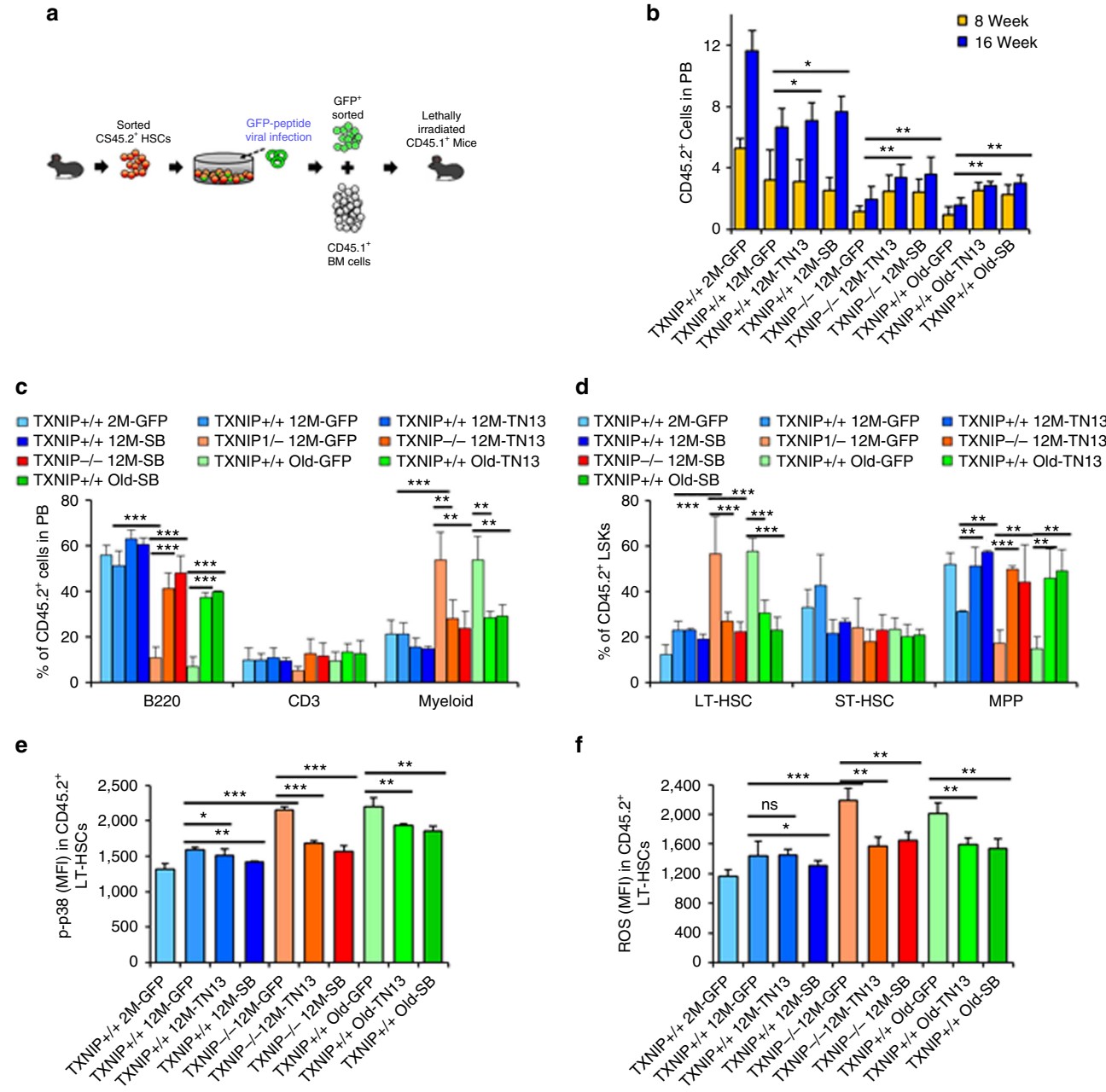

**Figure 5 | Rejuvenation of aged HSCs by GFP-TN13-expressing lentiviral transduction in vivo.** (**a**) Scheme of the experimental procedure for transplantation. Five hundred GFP[+] LT-HSCs were i.v. injected with competitor BM cells (CD45.1[+], $1.5 \times 10^6$) into lethally irradiated (9 Gy) CD45.1[+] congenic recipients. (**b**,**c**) Distribution of donor-derived WBCs in PB (**b**) and percentage of B220[+], CD3[+] and myeloid cells among donor-derived WBCs in PB (**c**) ($n = 8$–12). (**d**–**f**) LT-HSCs, ST-HSCs and MPPs among donor-derived LSKs in BM (**d**), levels of phospho-p38 (**e**) and levels of ROS in donor-derived LT-HSCs (**f**) ($n = 5$–7). Data are mean ± s.d. Statistical significance was determined using a two-tailed Student's t-tests. *$P < 0.05$, **$P < 0.01$, ***$P < 0.001$.

is induced by various pathological conditions or cellular ageing and results in HSC defects[1,4,6,40]. SB203580 administration restored the repopulation capacity, maintained the quiescence of HSCs and promoted the expansion of mouse and human HSCs ex vivo[1,25]. p38 was also activated in muscle stem cells from aged mice, and p38 inhibition has been shown to restore defects of muscle stem cells[41,42]. In human tissue-derived mesenchymal stem cells and endometrium-derived mesenchymal stem cells, p38 plays an important role in cellular senescence, and the pharmacological inhibition of p38 abrogated ageing phenotypes[43,44]. As noted above, the regulation of p38 activity appears to be a promising target for stem cell rejuvenation.

In this study, we analysed the functions of TXNIP on HSC ageing with $TXNIP^{-/-}$ mice that showed premature ageing phenotypes of HSCs. Ageing of $TXNIP^{-/-}$ HSCs was mostly due to the elevation of ROS and the induction of p38 activity. We identified the direct interaction between TXNIP and p38 via docking region of p38. We investigated the cellular function of p38 activity in $TXNIP^{-/-}$ HSCs and old HSCs using p38 chemical inhibitor and $TXNIP^{-/-}/p38^{AF/+}$ mice in vivo. From these data, we provided information that the activation of p38 in $TXNIP^{-/-}$ HSCs was major cause of HSC ageing and the inhibition of p38 activity in HSCs could rejuvenate the ageing phenotypes of aged HSCs.

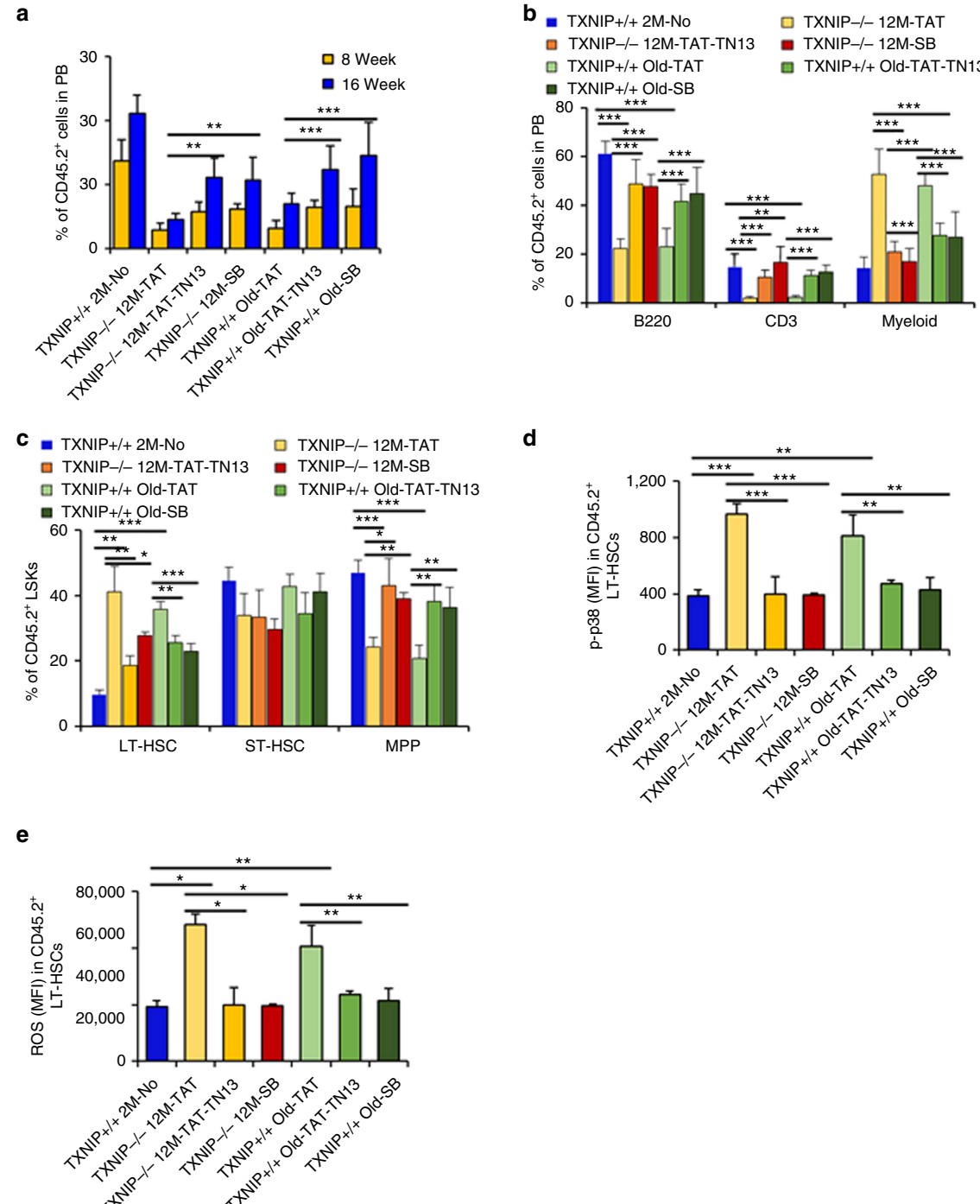

**Figure 6 | Rejuvenation of aged HSCs by TAT-TN13 *in vivo*. (a,b)** Distribution of donor-derived WBCs in PB (**a**) and percentage of B220[+], CD3[+] and myeloid cells among donor-derived WBCs in PB (**b**). Five hundred LT-HSCs were i.v. injected with competitor BM cells (*CD45.1*[+], $1.5 \times 10^6$) into lethally irradiated (9 Gy) *CD45.1*[+] congenic recipients (*n* = 8–13). (**c–e**) LT-HSCs, ST-HSCs and MPPs among donor-derived LSKs in BM (**c**) (*n* = 5–6), levels of phospho-p38 (**d**) and levels of ROS in donor-derived LT-HSCs (**e**) (*n* = 5). Data are mean ± s.d. Statistical significance was determined using a two-tailed Student's *t*-tests. *$P < 0.05$, **$P < 0.01$, ***$P < 0.001$.

The HSC ageing is initiated by DNA damage such as deletions and mutation, epigenetic alterations and altered expression of certain key transcription factors. Deregulation of these intrinsic factors drive HSCs into physiological ageing with combinatorial effects of these alterations. This implies HSC ageing seems to be reversible and can be reprogrammed by modulating these key factors. The rejuvenation of aged HSCs is the reversal of these alterations and functional restoration of aged HSCs. Several

approaches were tried and showed the evidences of at least partial HSC rejuvenation. Cdc42 inhibition restored the level of H4K16 acetylation of aged HSCs to that of young HSCs[4]. The overexpression of Sirt3 in aged HSCs decreased ROS level with partial recovery of all lineages[15]. mTOR inhibitor, rapamycin-reduced HSC numbers and increased reconstitution potential with balanced haematopoietic precursors[45,46]. Besides intrinsic factors, prolonged fasting restored frequency of myeloid-biased

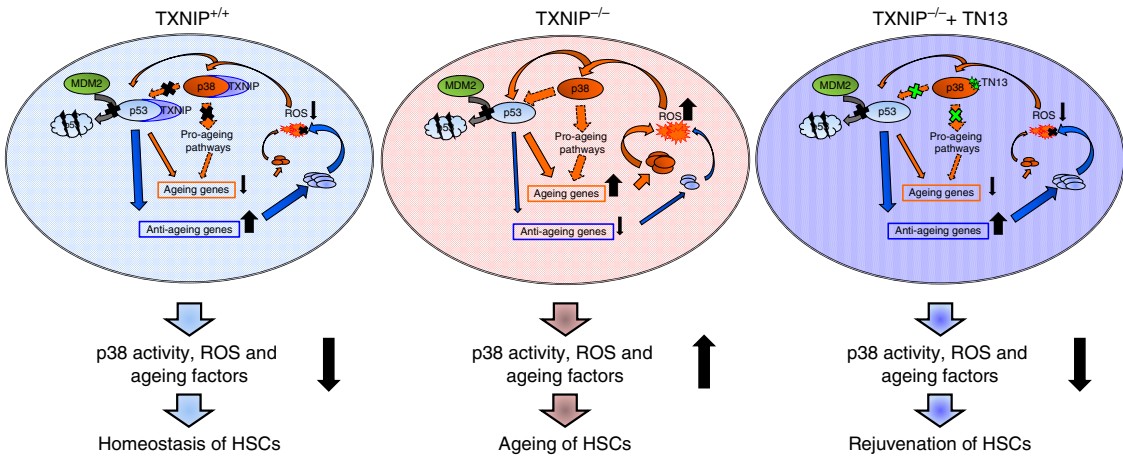

**Figure 7 | Regulation of ageing and p38 activity in HSCs by TXNIP or TXNIP-derived peptide.** A cartoon for the regulation of ageing of $TXNIP^{+/+}$ or $TXNIP^{-/-}$ HSCs by TXNIP or TAT-TN13. TXNIP-derived peptide, TAT-TN13, rejuvenates aged HSCs by inhibiting p38 activity, ROS production and ageing factors.

HSC population through reducing IGF-1 and protein kinase A activity[16]. The rejuvenation of HSCs will eventually improve and prevent the risks of ageing-associated diseases, organ dysfunction, malignancy and cancer.

Recently, many researchers have discussed the potential role of CPPs in the intracellular delivery and confirmed their efficiency *in vitro* and *in vivo*[47]. Especially, HIV TAT protein transduction domain was prominent in delivering efficiency in human and mouse haematopoietic cells[38,48–50]. And also some of groups proposed the TAT-conjugated peptide inhibitors for p38 and showed possible uses *in vitro* and *in vivo*[33,37]. Here, to develop our own therapeutic method to rejuvenate the aged HSCs, we designed CPP-conjugated peptide (TAT-TN13). TAT-TN13 inhibited the p38 activity efficiently via direct interaction and rejuvenated aged HSCs *in vitro*. Finally, we demonstrated the possibility of TAT-TN13 on rejuvenation of aged HSCs *in vivo* and TAT-TN13 had comparable effect to SB203580 and also rescued the old mice from 5-FU administration.

In conclusion, we demonstrated that TXNIP plays a crucial role in HSC ageing by inhibiting p38 activity via direct interaction. Overall, these data showed the possibility of TAT-TN13 derived from the docking motif of TXNIP as an applicable therapeutic drug for the rejuvenation of aged HSCs *in vivo*.

## Methods

**Mice.** $TXNIP^{-/-}$ mice (C57BL/6) were generated by homologous recombination, a targeting vector was constructed in which the sequence from exon one to exon eight was replaced with the *lacZ/neo* cassette gene[18] and congenic $CD45.1^+$ C57BL/6 mice and $p38^{AF/+}$ (B6.Cg-*Mapk14*$^{tm1.1Dvb}$/J) mice were obtained from The Jackson Laboratory. We crossed $TXNIP^{-/-}$ mice with $p38^{AF/+}$ mice. To confirm their genotypes, we used following primers, $TXNIP^{-/-}$ mice: forward, 5′-ATTCCCCTTCCAGGTGGA-3′, reverse, 5′-TTGAAATTGGCTCTGT-3′, LacZ: forward, 5′-GAAGCCAATATTGAAACCCA-3′, reverse, 5′-GCAAAGACCAGAC CGTTCAT-3′, $p38^{AF/+}$ mice: forward, 5′-TAGAGCCAGCCCCACTTTAGTC-3′, reverse, 5′-GAAGATGGATTTTAAGCATCCGT-3′.

Male mice were used for overall study and female mice were used only in Supplementary Fig. 1e as a control experiment. All mice were housed in a pathogen-free animal facility under a 12-h light-dark cycle. All animal experiments were approved by the Institutional Animal Use and Care Committee of the Korea Research Institute of Bioscience and Biotechnology and were performed in accordance with the Guide for the Care and Use of Laboratory Animals published by the US National Institutes of Health.

**Flow cytometry and cell isolation.** BM cells were isolated from mouse femurs, tibias, hipbones and shoulder bones by grinding tissues in RPMI 1640 medium (WelGENE, Daegu, Korea) plus 2% fetal bovine serum (FBS). RBCs were lysed using ACK buffer (0.15 M NH4Cl, 1.0 mM KHCO₃, 0.1 mM EDTA (pH 7.4)) or

not, and the cells were filtered through a strainer. BM cells were stained according to standard methods and were analysed using FACSCanto II (BD Biosciences) or sorted using FACSAria cell sorter (BD Biosciences). For flow cytometry analysis or cell sorting, we used following cell-surface markers, lineage markers: anti-CD11b-biotin (clone M1/70; 1:3,532 dilution), anti-Gr-1-biotin (clone RB6-8C5; 1:3,532 dilution), anti-B220-biotin (clone RA3-6B2; 1:2,247 dilution), anti-NK1.1-biotin (clone PK136; 1:297 dilution), anti-CD2-biotin (RM2-5; 1:445 dilution) and anti-TER119-biotin (1:297 dilution) were from BD biosciences and anti-Streptavidin-APC-eFluor780 (eBiosciences; 1:100 dilution), and anti-c-kit-PE (clone 2B8, BD biosciences; 1:100 dilution), anti-c-kit-PE-Cy7 (clone 2B8, BD biosciences; 1:100 dilution), anti-Sca-1-PE-Cy7/PE/BV421 (clone D7, BD biosciences; 1:100 dilution), anti-Sca-1-APC (clone D7, eBiosciences; 1:100 dilution), anti-CD34-FITC(1:50 dilution)/Alexa Fluor 647 (1:100 dilution)/PE (clone RAM34, BD biosciences; 1:100 dilution), anti-CD135-APC/BV421 (clone A2F10.1, BD biosciences; 1:100 dilution), anti-CD135-PE-Cy7(clone A2F10.1, eBiosciences; 1:100 dilution), anti-CD45.1 eFluor450 (clone A20, eBiosciences; 1:100 dilution), anti-CD45.2-V500 (clone 104, BD biosciences; 1:100 dilution), For PB analysis, anti-B220-PE (clone RA3-6B2, BD biosciences; 1:100 dilution), anti-CD3e-APC-efluor780 (clone 17A2, eBiosciences; 1:100 dilution), anti-CD3e-BV421/PE-Cy7 (clone 17A2, 145-2C11, BD biosciences; 1:100 dilution), anti-Gr-1-Alexa Fluor 488/eFluor 660 (clone RB6-8C5, eBiosciences; 1:100 dilution), anti-CD11b-PE-Cyanine7 (clone M1/70, eBiosciences; 1:100 dilution) and anti-CD45.2-APC (clone 104, BD biosciences; 1:100 dilution) were used. For Lineage⁻ c-kit⁺ cells preparation, we used MACS purification methods. For intracellular staining of TXNIP or phospho-p38, anti-TXNIP (clone D5F3E, Cell Signaling; 1:50 dilution), anti-rabbit IgG Alexa Fluor 647 (Life technology; 1:50 dilution) and anti-phospho-p38-APC (clone 4NIT4KK, eBiosciences; 1:40 dilution) were used.

**Competitive transplantation.** For competitive repopulation assays, LT-HSCs were isolated from young or old mice ($CD45.2^+$). 400-500 LT-HSCs were i.v. injected with competitor BM cells ($CD45.1^+$, $1.0 \times 10^6$ or $1.5 \times 10^6$) into lethally irradiated (9 Gy) $CD45.1^+$ congenic recipients (6–8 weeks old). The repopulation of donor-derived cells was monitored by staining PB from tail vein and BM cells with antibodies against indicated surface markers after 16 weeks.

**Limiting dilution analysis.** For limiting dilution assay, LT-HSCs were isolated from 12-month-old $TXNIP^{+/+}$ or $TXNIP^{-/-}$ mice ($CD45.2^+$). Serially diluted (500, 250, 125, 62.5 and 31.25) LT-HSCs were i.v. injected with competitor BM cells ($CD45.1^+$, $1.0 \times 10^6$) into lethally irradiated (9 Gy) $CD45.1^+$ congenic recipients (8-week-old). The repopulation of donor-derived cells was monitored by staining PB from tail vein and negative recipients were counted <1% in each lineage after 16 weeks.

**Short-term homing assay.** For short-term homing assay, freshly isolated HSCs were treated with 10 μM of TAT, TAT-TN13 (synthesized by Peptron, Korea) or SB203580 (S1076, Selleckchem) in HSC media (Myelocult M5300 (Stemcell Technologies) containing mSCF (250-03, PeproTech Korea) 100 ng ml⁻¹, mFlt3L (250-31 L, PeproTech Korea) 100 ng ml⁻¹ and mTPO (315-14, PeproTech Korea) 20 ng ml⁻¹) for 16 h at 37 °C, 5% CO₂. 10,000 $CD45.2^+$ HSCs were i.v. injected into lethally irradiated (9 Gy) $CD45.1^+$ congenic recipients. After 18 h, recipients were killed and BM cells were stained with anti-CD45.1 and CD45.2 antibodies for 25 min at 4 °C. $CD45.2^+$ cells were analysed by flow cytometry and relative frequency of $CD45.2^+$ cells were determined in total BM cells[4].

**Detection of ROS.** Levels of ROS in LT-HSCs were measured by flow cytometry using the ROS-specific fluorescent probe CM-DCF-DA (C6827, Molecular Probes/Thermofisher Scientific) and Dihydroethidium (DHE) (D11347, Molecular Probes/ThermoFisher Scientific). Freshly isolated BM cells were stained with surface markers and then incubated for 15 min at 37 °C with CM-DCF-DA at a final concentration of 5 μM or 2.5 μM (DHE) in phosphate-buffered saline (PBS) containing 2% FBS.

**Intracellular staining and confocal imaging.** For phosho-p38 and TXNIP analysis using flow cytometry, total BM cells were first stained with surface markers and then fixed and permeabilized with the Cytofix/Perm Solution (BD Biosciences). For TXNIP staining, the cells were blocked with PBS containing 5% BSA for 30 min at RT and then incubated with primary antibody (1:50 dilution) for 1 h in PBS containing 5% BSA, washed three times with PBS and then incubated with a secondary Alexa Fluor 647 (Life technology; 1:1,000 dilution) for 1 h at RT or for phospho-p38 staining, the cells were incubated with anti-phospho-p38-APC (1:40 dilution) at 4 °C for 30 min in PBS containing 2% FBS. For confocal imaging, freshly isolated LT-HSCs were seeded on fibronectin-coated coverslips at 4 °C for 10 min and were immediately fixed. After fixation, cells were permeabilized with 0.2% Triton X-100 for 20 min and blocked with PBS containing 5% BSA for 30 min at RT. Cells were stained primary and secondary antibodies for 1 h at RT and then were mounted with DAPI containing mounting reagent (Molecular Probes). For polarity staining, freshly isolated HSCs were seeded on fibronectin-coated coverslips at 4 °C for 10 min and then treated with 10 μM of TAT, TAT-TN13 or SB203580 in HSC media (Myelocult M5300 containing SCF 100 ng ml$^{-1}$, FLt3L 100 ng ml$^{-1}$ and TPO 20 ng ml$^{-1}$) for 16 h at 37 °C, 5% CO$_2$. After fixation, cells were permeabilized with 0.2% Triton X-100 for 20 min and blocked with PBS containing 5% BSA for 30 min at RT. Cells were stained anti-rabbit-Cdc42 (Cell Signaling; 1:200 dilution) and then stained with a secondary Alexa Fluor 546 (Life technology; 1:1,000 dilution) for 1 h at RT and then were mounted with DAPI containing mounting reagent (Molecular Probes). Fifty to sixty individual HSCs were scored for polarity of Cdc42. For *in situ* PLA assay, freshly isolated HSCs were seeded on fibronectin-coated coverslips at 4 °C for 10 min and treated with H$_2$O$_2$ or not in HSCs media for indicated time. Cells were fixed and permeabilized as described above. We examined *in situ* interaction between TXNIP and p38 using Duolink assay kit (Sigma) and we followed the manufacturer's instruction. In this assay, anti-mouse-TXNIP (K0205-3, MBL; 1:200 dilution) and anti-rabbit-p38 (9212, Cell Signaling; 1:200 dilution) were used. The images were captured using a LSM510 confocal microscope (Carl Zeiss).

**Recombinant constructs.** For GST pull-down assay, we constructed *TXNIP* and *p38α* mutants using site-directed mutagenesis. Human *TXNIP* or *p38α* clone was used as a template. For *TXNIP* mutants construction, we used the primers as follows: *TXNIP*(V178A/V180A), forward 5′-CCT GAT GGG CGG GCG TCT GCC TCT GCT CGA ATT-3′ and reverse 5′-AAT TCG AGC AGA GGC AGA CGC CCG CCC ATC AGG-3′; *TXNIP*(I208A/V210A), forward 5′-ACA TGT TCC CGA GCT GTG GCC GCC-3′ and reverse 5′-GGC AGC TTT GGG GGC CAC AGC TCG GGA ACA TGT-3′; *TXNIP*(I278A/V280A), forward 5′-TAT TCC TTA CTG GCC TAT GCT AGC GTT CCT GGA-3′ and reverse 5′-TCC AGG AAC GCT AGC ATA GGC CAG TAA GGA ATA-3′; *TXNIP*(L290A/L292A), forward 5′-AAG AAG GTC ATC GCT GAC GCG CCC CTG GTA ATT-3′ and reverse 5′-AT TAC CAG GGG CGC GTC AGC GAT GAC CTT CTT-3′; *TXNIP*(K286A/K287A/L290A/L292A), forward 5′-GCG GCG GTC ATC GCT GAC GCG CCC CTG GTA ATT-3′ and reverse 5′-AAT TAC CAG GGG CGC GTC AGC GAT GAC CGC CGC-3′; *p38*(I116A), forward 5′-GAT CTG AAC AAC GCT GTG AAA TGT CAG-3′ and reverse 5′-CTG ACA TTT CAC GCG GTT GTT CAG ATC-3′; *p38*(Q120A), forward 5′-ATT GTG AAA TGT GCG AAG CTT ACA GAT-3′ and reverse 5′-ATC TGT AAG CTT CGC ACA TTT CAC AAT-3′ *p38*(E160A/D161A), forward 5′-CTA GCT GTG AAT GCA GCC TGT GAG CTG AAG-3′ and reverse 5′-CTT CAG CTC ACA GGC TGC ATT CAC AGC TAG-3′ *p38*(D313A/D315A/D316A), forward 5′-GCT CAG TAC CAC GCT CCT GCT GCT GCA CCA GTG GCC-3′ and reverse 5′-GGC CAC TGG TTC AGC AGC AGG AGC GTG GTA CTG AGC-3′, *p38AF*, forward 5′-GAT GAA ATG GCA GGC TTC GTG GCC ACT-3′ and reverse 5′-AGT GGC CAC GAA GCC TGC CAT TTC ATC-3′. To construct lentiviral clones, we used pLVX-AcGFP1-C1 vector. To use internal BamH1 restriction site of TXNIP interaction motif, we mutated BamH1 site of pLVX-AcGFP1-C1 vector using Bgl2 site of these double-stranded oligoes: forward 5′-GC GAA TTC TGC AGT CGA CGG TAC CGC GGG CCC GAG ATC T GC-3′ and reverse 5′-C AGA TCT CGG GCC CGC GGT ACC GTC GAC TGC AGA ATT C GC-3′ and each peptide oligoes cloned into Xho1 and EcoR1 restriction sites of pLVX-AcGFP1-C1 vector. Then, we inserted 5X linker sequence into Xho1 and BamH1 sites of pLVX-AcGFP1-C1-peptide vector. 5X linker oligoes have following sequences: forward 5′-gc ctc gag ct GAA GCT GCT GCT AAA GAA GCT GCT GCT AAA GAA GCT GCT GCT AAA GAA GCT GCT GCT AAA GAA GCT GCT GCT gga tcc gc-3′, reverse 5′-gc gga tcc AGC AGC AGC TTT AGC AGC AGC TTC TTT AGC AGC AGC TTC TTT AGC AGC AGC TTC TTT AGC AGC AGC TTC ag ctc gag gc-3′. Primers were synthesized by Bioneer (Korea).

**In vitro kinase assay.** GST, GST-TXNIP (150–317) and His-p38α were purified using affinity chromatography. To examine the kinase activity of p38 *in vitro*, we used p38 MAP Kinase assay kit (9820, Cell Signaling). We added 0.5–1 μg of His-p38α for kinase assay and its activity was determined by phospho-ATF2 levels. We obtained clones (human p38α, β, γ and δ) from Korea Human Gene Bank, Medical Genomics Research Center, KRIBB, Korea. For *p38* isoform construction, we used the primers as follows: *p38α*, forward 5′-gcctgcagatgtctcaggagaggcccacg-3′ and reverse 5′-gctctagatcaggactccatctctcttggtc-3′; *p38β*, forward 5′-gcgaattcatgtc gggccctcg cgccggc-3′ and reverse 5′-gcgtcgactcactgctcaatctccaggct-3′; *p38γ*, forward 5′-gcgaattcatgagctctccgccgcccgcc-3′ and reverse 5′-gcgtcgactcacagaggcgtctccttgga-3′; *p38δ*, forward 5′-gcgaattcatgagcctcatccggaaaaag-3′ and reverse 5′-gcgtcgacctacag cttcatgccact ccg-3′, and we cloned into FLAG-CMV vector using these restriction sites, *p38α* (Pst1/Xba1), *p38β*(EcoR1/Sal1), *p38γ*(EcoR1/Sal1) and *p38δ*(EcoR1/Sal1) for mammalian cell expression. FLAG-p38α, β, γ and δ were expressed in 293T cells (ATCC). p38 isoforms were immunoprecipitated with α-FLAG antibody (Sigma; 1:250 dilution) then we performed *in vitro* kinase assay. The uncropped western blots can be found in Supplementary Fig. 8.

**ITC assay.** For ITC assay, all measurements were carried out at 25 °C on a microcalorimetry system iTC200 (GE Healthcare). p38α and peptides were dialysed against a solution containing PBS. The samples were centrifuged to remove any residuals before the measurements. The experiments were carried out by titrating 1,050 μM TAT-TN13 peptide into 57.99 μM p38α protein. Dilution enthalpies were determined in separate experiments (titrant into buffer) and subtracted from the enthalpies of the binding between the proteins. Data were analysed using the Origin software (OriginLab).

**Quantitative real-time PCR.** The total cellular RNA was isolated from 40,000–50,000 LT-HSCs using RNeasy Micro kit (Qiagen). Quantitative real-time PCR was performed using the SYBR Premix ExTaq (Takara Bio) and a Thermal Cycler Dice Real-Time System TP800 instrument (Takara Bio). We used following primers: *p16*, forward 5′-CGAACTCTTTCGGTCGTACC-3′ and reverse 5′-CGA ATCTGCACCGTAGTTGA-3′, *p19*, forward 5′-GCTCTGGCTTTCGTGAA CAT-3′ and reverse 5′-TCGAATCTGCACCGTAGTTGA-3′, *p21*, forward 5′-CTG TCTTGCACTCTGGTGTC-3′ and reverse 5′-CCAATCTGCGCTTGGAGTGA-3′, *Wnt5a*, forward 5′-CTCTAGCGTCCACGAACTCC-3′ and reverse 5′-CAAATAG GCAGCCGAGAGAC-3′ and *β-actin*, forward 5′-CTCCTGAGCGCAAGTAC TCT-3′ and reverse 5′-TAAACGCAGCTCAGTAACA-3′.

**Statistical analysis.** The data are expressed as the mean ± s.d. of n determinations and statistical significance was determined using a two-tailed Student's t-tests. *$P < 0.05$, **$P < 0.01$, ***$P < 0.001$.

**Data availability.** The authors declare that all data supporting the findings of this study are available within the article and its Supplementary Information files or from the corresponding author upon reasonable request.

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

## Acknowledgements
We thank Hyang Ran Yoon (KRIBB) for experimental help. This work was supported in part by research funding from the R&D convergence Program (No CRC-15-02-KRIBB) of NST, the Ministry of Science, ICT & Future Planning, and the Korean Health Technology R&D Project (A121934), Ministry of Health and Welfare, and KRIBB Research Initiative Program, Republic of Korea.

## Author contributions
H.J. and D.O.K. designed and performed experiments, analysed the data and wrote the manuscript. J.-E.B., W.S.K., M.J.K. and H.Y.S. performed mice experiments. Y.K.K. and D.-K.K. purified recombinant proteins and performed ITC assay. Y.-J.P., T.-D.K., S.R.Y. and H.G.L. provided helpful discussions and crucial analysis of data. E.-J.C. and S.-H.M. performed flow cytometry analysis, plasmid construction and immunoprecipitation assays. I.C. supervised the overall project, analysed the data and wrote the manuscript.

## Additional information

**Competing financial interests:** The authors declare no competing financial interests.

