## [Peer Review File · Nature Communications]

Reviewers' comments:

Reviewer #1 (Remarks to the Author):

General Comments

The authors present new information of interest regarding the role of Thioredoxin-interacting protein and p38 in the regulation of aged vs. younger hematopoietic stem cells (HSC). This is an in-depth analysis whose significance is heightened by multiple means to understand and to obviate the aging phenotype of HSC seen in older +/+ mice and in younger (12 month old) TXNIP^{-/-} mice. However, clarification is needed in some areas, more in-depth competitive transplant studies are needed, and some of the statements should be "toned" down a bit.

Specific Comments (Major)

1. The competitive engraftment studies are of interest, but data is only shown for one time point. It will be important to show the engraftment chimerism over time for the blood, at least for CD45.2+ cell engraftment. Most importantly, at least for some of the engraftment studies the investigators need to do limiting dilution analysis of donor cells to calculate the competitive repopulating units (CRUs). Only in this way can one be sure about the numbers of functional HSC.
2. Be more precise when mentioning old vs. young for all Figures unless you clearly define in the text that old equals about 24 months and younger (12 months, or whatever this is). Also, are the results seen similar for male and female mice? What were the sexes of the donors and the recipients?
3. It is not clear what any of the 5FU studies prove. The responses to 5FU could be due to many factors, including whether there are more HSC and progenitors in cycle, and hence the mice with enhanced cell cycling would be more sensitive to cell kill by 5FU. I suggest removing all the data on 5FU effects, unless a better case can be made for including these results in the paper.
4. Page 5, 5 lines from bottom and Fig. 1C: Show the comparative 24 month old +/+ data here for comparison with the 12 month old data
5. In the Methods Section, you mention short-term homing. I may have missed it, but where is this data or mention of homing in the text? This is important information so does need to be included if it is not already.

Minor Comments

1. Some of the statements should be "toned" down. For example:
 - a. Page 6, lines 12-13: "These data indicate that TXNIP **may** play a role...". Also, remove word pivotal.
 - b. Page 6, lines 12-14: "...loss of TXNIP **may** induce premature ageing..."
 - c. Page 7, line 5: "...TXNIP and the p38 pathway **may** regulate..."
 - d. Page 7: Mention in text what SB203580 is so readers don't need to look elsewhere for this information.

- e. Page 10, line 15: "...our data **suggests** that...". Remove words strongly implied.
- f. Figure 7 is very useful but rather than as a separate figure you might consider placing this as a subfigure for figure 6.
- g. Page 13, first line of second paragraph: "A **potentially** promising target..."
- h. Page 13, 2 lines from bottom: "From these data we **provided** information that links the activation...", rather than saying we proved that.

Reviewer #2 (Remarks to the Author):

The authors' previous publications showed that TXNIP was expressed at the highest levels in LT-HSCs and its expression decreased when the cells were activated to undergo proliferation and differentiation. TXNIP KO mice exhibited defects in hematopoietic cells, particularly in the HSC population, including reduced frequencies of HSCs and HPCs in the bone marrow, a decreased ability of HSCs to generate long-term engraftment after competitive transplantation, and a higher sensitivity of HSCs to oxidative stress. These defects were attributed to a dysregulation of p53 activity and increased production of ROS in TXNIP KO HSCs. The results reported in the current manuscript represent an incremental progression of their previous works by demonstrating that activation of p38 induced by oxidative stress also played an important role in the defects of TXNIP KO HSCs. However, the roles of ROS and p38 activation in HSC aging have been well established.

Concerns:

1. In the authors' previous publications (J Immunol, 2009 & Cell Metabolism, 2013), they showed that TXNIP KO mice at 12 and 24 months had significantly lower frequencies of LT-HSCs and CD150+CD48-LSK cells in the bone marrow compared to age-matched WT mice. However, in the present manuscript, the authors showed that TXNIP KO mice at 12 months had significantly higher frequencies of LT-HSCs, SH-HSCs and MPPs in LSK cells than age-matched WT mice (Fig. 1b). To make the data more consistent, the authors should present their data as percentages of bone marrow cells and absolute number of LT-HSCs, SH-HSCs and MPPs in each mouse.
2. In the authors' previous publications and current manuscript, they hypothesized that increased production of ROS is the primary cause of HSC defects in TXNIP KO mice. This should be tested by treating the KO mice with an antioxidant to see if antioxidant can inhibit p38 activation and defects of HSCs in TXNIP KO mice.
3. Does increased production of ROS contribute to the higher sensitivity of TXNIP KO HSCs to 5-FU? Can antioxidant treatment reduce their sensitivity?
4. The levels of ROS production in donor-derived LT-HSCs from TXNIP KO mice presented in Fig. 3e were significantly higher than that presented in Fig. 1i. Please clarify the discrepancy.
5. In Fig. 3e, the authors showed that inhibition of p38 activation reduced ROS production by donor-derived LT-HSCs from TXNIP KO mice. Please elaborate why p38 inhibition can reduce ROS production in LT-HSCs.
6. Since inhibition of p38 with a small molecule can also rejuvenate aged tissue stem cells, what is the advantage of using TAT-TN13 to inhibit p38 for rejuvenate aged HSCs?
7. The authors should discuss their previous findings regarding the mechanisms by which TXNIP KO causes defects in HSCs in the discussion and present a better model to summarize their finding in Fig. 7.

Reviewer #3 (Remarks to the Author):

Abstract

The manuscript describes a study of one of the molecular mechanisms of hematopoietic stem cell (HSC) aging and the potential to rejuvenate HSCs. The author has demonstrated that thioredoxin-interacting protein (TXNIP) regulates the aging of HSCs by inhibiting p38 activity via direct interactions in HSCs. In addition, through similar interacting mechanisms to inhibit p38 activity, the author has derived cell penetrating peptide (CPP)-conjugated peptide (TAT-TN13) from the TXNIP-p38 interaction motif, and showed that the peptide is able to rejuvenate aged HSCs both in vitro and in vivo.

The strengths of the study include the investigation of an important question regarding the aging of HSCs that might lead to aging-associated diseases in most elderly people. The findings of this study suggest a novel regulatory mechanism via TXNIP-p38 axis in HSC aging and developed a therapeutic method to rejuvenate aged HSCs. Such findings might be of interest to readers in the field of aging studies and stem cell therapy.

Some specific areas to attend to are stated as below.

Major concerns:

1. The author demonstrated the role of TAT-TN13 in altering mRNA level of several aging associated genes in vitro, will such observations exhibit similar patterns after competitive transplantation assay in vivo?
2. Regarding the design of GFP-TN13 rejuvenating HSC experiments (Figure 5).
 - (1). 6 or 12 month old mice should be included, as 2 month old mice are still considered young for aging studies.
 - (2). It would be better to include SB203580 administration comparison in GFP-TN13 studies as well.
3. The results demonstrate that GFP-TN13 and TAT-TN13 inhibit p38 activity in a very similar manner with SB203580. Are they specific to p38alpha?

What is the advantage to use the newly derived peptide over the traditional p38 inhibitor in term of dosage, efficiency and specificity?

4. How is the stability of TAT-TN13 peptide in vivo? In order to maintain the rejuvenating effect, what should be the frequency of administration for long term effect?
5. Recent study suggests that aging is a multifactorial process, so that modulation of aging with a compound to rejuvenate one organ could potentially lead to adverse outcomes in others (Wahlestedt M, 2015). Has the author observed any adverse effect in other organs of the mice after the injection of TXNIP derived peptide? How is the dosage determined?

Minor concerns

1. Figure 2D seems redundant as it is also shown in part of Figure 2e.
2. What is the purpose of introducing GFP-TN13 to rejuvenate HSC besides TAT-TN13? How is the effect compared to TAT-TN13 in parallel?
3. In Figure 7, in the aged HSC, the statement that 'a loss of TXNIP resulted in p38 activation in HSCs' is not clearly depicted in the illustration, as the number of TXNIP should decrease in aged HSCs that results in an increase of p38 activity.

References

Wahlestedt M, Pronk CJ, Bryder D, (2015), Concise review: hematopoietic stem cell aging and the prospects for rejuvenation. *Stem Cells Transl Med.* 4(2):186-94. doi: 10.5966/sctm.2014-0132.

Point-to-point response

Reviewer #1

General Comments

The authors present new information of interest regarding the role of Thioredoxin-interacting protein and p38 in the regulation of aged vs. younger hematopoietic stem cells (HSC). This is an in-depth analysis whose significance is heightened by multiple means to understand and to obviate the aging phenotype of HSC seen in older +/+ mice and in younger (12 month old) TXNIP^{-/-} mice. However, clarification is needed in some areas, more in-depth competitive transplant studies are needed, and some of the statements should be "toned" down a bit.

Specific Comments (Major)

1. The competitive engraftment studies are of interest, but data is only shown for one time point. It will be important to show the engraftment chimerism over time for the blood, at least for CD45.2⁺ cell engraftment. Most importantly, at least for some of the engraftment studies the investigators need to do limiting dilution analysis of donor cells to calculate the competitive repopulating units (CRUs). Only in this way can one be sure about the numbers of functional HSC.

Response:

We appreciate the reviewer's valuable comments. As mentioned by the reviewer, we have revised our data and text.

1. We have 8-week point chimerism data for all competitive transplantation assays but our experiments have so many experimental sets, so we have tried to simplify the data at 16-week point engraftment chimerism. However, as you commented, we added the new data at 8-week point for the CD45.2⁺ engraftment in our **new version Fig. 1h, Fig. 3a, Fig. 5b and Fig. 6a.**
2. We have done a limiting dilution analysis of donor cells to calculate the competitive repopulating units (CRUs) from 12-month-old TXNIP^{+/+} and TXNIP^{-/-} mice and

added figures (our new version **Figure 11**), described the results in the text (**page 6, line 23- page 7, line 3**). The detailed methods were described in Methods section (**page 18**). As expected, 12-month-old KO HSCs have shown markedly reduced numbers of functional HSCs.

2. Be more precise when mentioning old vs. young for all Figures unless you clearly define in the text that old equals about 24 months and younger (12 months, or whatever this is). Also, are the results seen similar for male and female mice? What were the sexes of the donors and the recipients?

Response:

1. In our experiments, “young” indicates 2-month-old mouse and “old” indicates 24-month-old mouse and we have described the age of mice in the text page5, line 7 to differentiate their age “at 2 (young), 6, 12 and 24 (old) months”.
2. We have missed the sexes of mice in our text or Methods section, in this study. We used male mice for donors, recipients and other analyses. It was described in our new version **Methods section (page 17, line 10-11)**.
3. Additionally, we investigated the ratio of white blood cells (WBCs) in the PB of 12-month-old TXNIP^{+/+} and TXNIP^{-/-} female mice to confirm the ageing phenotype of female mice. 12-month-old TXNIP^{-/-} female mice showed markedly skewed differentiation to myeloid as shown in 12-month-old TXNIP^{-/-} male mice (**new version Supplementary Figure 1e**)

3. It is not clear what any of the 5FU studies prove. The responses to 5FU could be due to many factors, including whether there are more HSC and progenitors in cycle, and hence the mice with enhanced cell cycling would be more sensitive to cell kill by 5FU. I suggest removing all the data on 5FU effects, unless a better case can be made for including these results in the paper.

Response:

1. Reviewer 2 also has raised some comments for 5-FU studies therefore, we performed some 5-FU related experiments and then we moved all of 5-FU related figures to

supplementary figures (**new Supplementary Fig. 1g, Fig. 1h, Fig. 3, Fig. 7**).

2. To examine whether the increased ROS production contributes to the higher sensitivity of TXNIP^{-/-} HSCs to 5-FU treatment, we administered 5-FU with NAC (N-acetyl-L-cysteine), an antioxidant agent. As shown in new version **Supplementary Fig. 1g and Fig. 1h**, TXNIP^{-/-} HSCs were more sensitive than TXNIP^{+/+} HSCs to 5-FU treatment and mice survival was fully rescued by NAC treatment. These results suggest that the increased production of ROS in TXNIP^{-/-} HSCs may result in the defects in the repopulation capacity of HSCs under haematopoietic stress. We also described the details in the text (**page 6, line 3- 13**).

4. Page 5, 5 lines from bottom and Fig. 1C: Show the comparative 24 month old +/+ data here for comparison with the 12 month old data

Response:

We have added the ROS levels in 24 month old TXNIP^{+/+} HSCs for comparison with the 12 month old data (**new version Fig. 1c**)

5. In the Methods Section, you mention short-term homing. I may have missed it, but where is this data or mention of homing in the text? This is important information so does need to be included if it is not already.

Response:

1. Reviewer might have missed it. You can find the data in our **Figure 4I** and we have described it in **page 12, line 7-8**.

Minor Comments

1. Some of the statements should be "toned" down. For example:

- a. Page 6, lines 12-13: "These data indicate that TXNIP **may** play a role...". Also, remove word pivotal.

Response:

We have changed the text as reviewer mentioned.

b. Page 6, lines 12-14: "...loss of TXNIP **may** induce premature ageing..."

Response :

We have changed the text as reviewer mentioned.

c. Page 7, line 5: "...TXNIP and the p38 pathway **may** regulate..."

Response :

We have changed the text as reviewer mentioned.

d. Page 7: Mention in text what SB203580 is so readers don't need to look elsewhere for this information.

Response :

We have already mentioned it in the text **page 4, line 4** ("SB203580, a p38 inhibitor")

e. Page 10, line 15: "...our data **suggests** that...". Remove words strongly implied.

Response :

We have changed the text as reviewer mentioned.

f. Figure 7 is very useful but rather than as a separate figure you might consider placing this as a subfigure for figure 6.

Reponse :

Reviewer 2 and 3 also have raised comments for the illustration of TXNIP functions in HSCs. We have illustrated the functions of TXNIP and TN13 in HSCs and placed in **new version Fig. 7**.

g. Page 13, first line of second paragraph: "A **potentially** promising target..."

Response :

We have added “ potentially” in the text.

h. Page 13, 2 lines from bottom: "From these data we **provided information that links** the activation...", rather than saying we proved that.

Response :

We have changed the text as reviewer mentioned.

Reviewer #2:

The authors' previous publications showed that TXNIP was expressed at the highest levels in LT-HSCs and its expression decreased when the cells were activated to undergo proliferation and differentiation. TXNIP KO mice exhibited defects in hematopoietic cells, particularly in the HSC population, including reduced frequencies of HSCs and HPCs in the bone marrow, a decreased ability of HSCs to generate long-term engraftment after competitive transplantation, and a higher sensitivity of HSCs to oxidative stress. These defects were attributed to a dysregulation of p53 activity and increased production of ROS in TXNIP KO HSCs. The results reported in the current manuscript represent an incremental progression of their previous works by demonstrating that activation of p38 induced by oxidative stress also played an important role in the defects of TXNIP KO HSCs. However, the roles of ROS and p38 activation in HSC aging have been well established.

Concerns:

1. In the authors' previous publications (J Immunol, 2009 & Cell Metabolism, 2013), they showed that TXNIP KO mice at 12 and 24 months had significantly lower frequencies of LT-HSCs and CD150+CD48-LSK cells in the bone marrow compared to age-matched WT mice. However, in the present manuscript, the authors showed that TXNIP KO mice at 12 months

had significantly higher frequencies of LT-HSCs, SH-HSCs and MPPs in LSK cells than age-matched WT mice (Fig. 1b). To make the data more consistent, the authors should present their data as percentages of bone marrow cells and absolute number of LT-HSCs, SH-HSCs and MPPs in each mouse.

Response :

Thank you for your valuable comments. We calculated the percentages of bone marrow cells and absolute number of LT-HSCs, SH-HSCs and MPPs, and presented the results in our new version **Supplementary Fig. 1c,d**. As shown in figures, TXNIP^{-/-} mice showed similar results with our previous results. These data showed the exhaustion of HSCs in TXNIP^{-/-} mice by high ROS. We have added this text **“Next, we analyzed the frequency of BM cells and absolute number of LT-HSCs, ST-HSCs and MPPs in each mouse. TXNIP^{-/-} mice showed the exhaustion of HSCs at 12- and 22-month age (Supplementary Fig. 1c,d).”** into our new version text **page 5, line 12-14**.

2. In the authors' previous publications and current manuscript, they hypothesized that increased production of ROS is the primary cause of HSC defects in TXNIP KO mice. This should be tested by treating the KO mice with an antioxidant to see if antioxidant can inhibit p38 activation and defects of HSCs in TXNIP KO mice.

Response:

To test the efficiency of an antioxidant on ROS production and p38 activation in TXNIP^{-/-} HSCs, we have treated NAC (N-acetyl-L-cysteine/ 100 mg/kg) to TXNIP^{+/+} 2-month-old mice or TXNIP^{-/-} 12-month-old mice for 12 hours. Then, we investigated the levels of ROS and p38 activation using flow cytometry. As shown in new version **Supplementary Fig. 2a,b**, ROS levels and p38 activation were decreased by a single injection of NAC (**page 7, line 12-14**).

3. Does increased production of ROS contribute to the higher sensitivity of TXNIP KO HSCs to 5-FU? Can antioxidant treatment reduce their sensitivity?

Response:

Reviewer 1 also has raised 5-FU related comments. We administered 5-fluorouracil (5-FU) to induce a transient leukopenia in the blood with NAC (N-acetyl-L-cysteine) to examine whether the increased ROS production contributes to the higher sensitivity of TXNIP^{-/-} HSCs to 5-FU treatment. As shown in **new version Supplementary Fig. 1g,h**, TXNIP^{-/-} HSCs were more sensitive than TXNIP^{+/+} HSCs to 5-FU treatment and mice survival was fully rescued by NAC treatment. These results suggest that the increased production of ROS in TXNIP^{-/-} HSCs may result in the defects in the repopulation capacity of HSCs under haematopoietic stress. We also have described the details in the new version text (**page 6, line 3- 13**).

4. The levels of ROS production in donor-derived LT-HSCs from TXNIP KO mice presented in Fig. 3e were significantly higher than that presented in Fig. 1i. Please clarify the discrepancy.

Response:

The relative ROS production may be determined by many factors. The sensitivity of detection is depending on the resetting of flow cytometry. Another possible factor is the sensitivity of ROS reagents. The ROS-specific fluorescent probe CM-DCF-DA is very sensitive to reaction time and storing condition. There are time gaps between incubation. In Figure 3e, we have so many experimental sets, so the reaction time might be delayed. As you know, the MFI levels are not the absolute levels of ROS. When we examined the ROS levels using flow cytometry, their levels were flexible. Therefore, it is reasonable to compare their relative levels of ROS in each experimental set.

5. In Fig. 3e, the authors showed that inhibition of p38 activation reduced ROS production by donor-derived LT-HSCs from TXNIP KO mice. Please elaborate why p38 inhibition can reduce ROS production in LT-HSCs.

Response:

As described in our text, many researchers have observed p38 activation in various

pathological conditions or during cellular ageing via elevated ROS, resulting in HSC defects. Activation of p38 can induce pro-oxidant pathways from oxidative stress or other stimuli and also activated p38 can induce ROS production during stress-induced signaling pathways. As shown in **below reference figures**, in various conditions, p38 activation induces ROS production in cells and p53 also can be activated by p38 to induce ageing factors. In our **new version Figure 7**, we have illustrated the possible mechanism of ROS regulation in HSCs by p38, p53, TXNIP or p38 inhibitor (TN13).

Reference figure 1. from *Blood*. 2011 Jun 2;117(22):5953-62.

Reference figure 2. from *Am J Physiol Heart Circ Physiol* 296: H470–H479, 2009.

Reference figure 3. from *Biochem. J.* (2007) 402, 271–278

Reference figure 4. from *Nature Reviews Microbiology* (2013) 11,615–626.

Reference figure 5. from *TRENDS in Biochemical Sciences* Vol.32 No.8, 2007

6. Since inhibition of p38 with a small molecule can also rejuvenate aged tissue stem cells,

what is the advantage of using TAT-TN13 to inhibit p38 for rejuvenate aged HSCs?

Response:

As shown in our overall results, TN13 is comparable to SB203580 in p38 activity inhibition and rejuvenation of HSCs. TN13 is derived from TXNIP binding sequence of p38 docking site but small molecule inhibitors like SB203580 are competitors of ATP binding site of p38 kinase.

Recent studies have reported the side effects of small molecule inhibitors of p38 from nonspecific targeting (Nat Biotechnol. 2008 Jan;26(1):127-32., J Biol Chem. 2000 Mar 10;275(10):7395-402., British Journal of Pharmacology (2000) 131, 99 – 107). To inhibit signaling pathways in cells, the specific targeting is the most important point. To determine the specificity of TN13, we tried kinase assays for p38 isoforms *in vitro*. As shown in new version **Supplementary Fig. 4i**, SB203580 completely inhibited 2 isoforms, p38 α and p38 β but TN13 showed the specificity on p38 α (**page 10, line 21- page 11, line 1**). It suggests that TN13 may more specific than SB203580 for p38 α inhibition and we may reduce the side effects *in vitro* and *in vivo* by using TN13 as a p38 α inhibitor. And also TN13 is water soluble but SB203580 is DMSO soluble. Drug solubility is also important factor for the application to cells or clinical trials to avoid cytotoxicity in high concentration. We are going to test the side effects of TN13 *in vivo* and *in vitro* further including kinase profiling and toxicity assays.

7. The authors should discuss their previous findings regarding the mechanisms by which TXNIP KO causes defects in HSCs in the discussion and present a better model to summarize their finding in Fig. 7.

Response:

Reviewer 1 and 3 also commented on the illustration of TXNIP functions in HSCs. We have illustrated the functions of TXNIP and TN13 in HSCs and revised it in **new version Fig. 7**. As shown in **new version Fig. 7**, TXNIP inhibits p38 activity and activates p53 via direct interaction and these interactions are resulted in the reduction of aging-associated gene expression or the induction of p53 dependent anti-oxidant genes. As a result, TXNIP^{+/+} HSCs can maintain low level of ROS and their

homeostasis. However, in TXNIP^{-/-} HSCs, the loss of TXNIP makes them free and ROS-inducing circuit will be turned on easily. At this condition, unregulated ROS may hyper-activate p38 or p53 that induces ageing factors and make HSCs to become old. These ageing phenotypes can be reversed by p38 inhibition via TN13 treatment.

Figure 7. Regulation of ROS in HSCs by TXNIP or TXNIP-derived peptide.

Reviewer #3:

Abstract

The manuscript describes a study of one of the molecular mechanisms of hematopoietic stem cell (HSC) aging and the potential to rejuvenate HSCs. The author has demonstrated that thioredoxin-interacting protein (TXNIP) regulates the aging of HSCs by inhibiting p38 activity via direct interactions in HSCs. In addition, through similar interacting mechanisms to inhibit p38 activity, the author has derived cell penetrating peptide (CPP)-conjugated peptide (TAT-TN13) from the TXNIP-p38 interaction motif, and showed that the peptide is able to rejuvenate aged HSCs both in vitro and in vivo.

The strengths of the study include the investigation of an important question regarding the aging of HSCs that might lead to aging-associated diseases in most elderly people. The findings of this study suggest a novel regulatory mechanism via TXNIP-p38 axis in HSC aging and developed a therapeutic method to rejuvenate aged HSCs. Such findings might be of interest to readers in the field of aging studies and stem cell therapy.

Some specific areas to attend to are stated as below.

Major concerns:

1. The author demonstrated the role of TAT-TN13 in altering mRNA level of several aging associated genes *in vitro*, will such observations exhibit similar patterns after competitive transplantation assay *in vivo*?

Response:

We have isolated CD45.2⁺ HSCs from competitive transplanted recipients after 16 weeks. We investigated the mRNA level of ageing associated genes *in vivo*. As shown in **below figure**, alterations of mRNA level were similar with *in vitro* data in **Fig. 4h-k** and we have added these results as a **new version Supplementary Fig. 6 (page 12, line 4-6)**.

Supplementary Figure 6. Long-term regulation of ageing-associated genes in HSCs by TAT-TN13 treatment *in vivo*.

2. Regarding the design of GFP-TN13 rejuvenating HSC experiments (Figure 5).

(1). 6 or 12 month old mice should be included, as 2 month old mice are still considered young for aging studies.

(2). It would be better to include SB203580 administration comparison in GFP-TN13 studies as well.

Response:

1. As the reviewer mentioned, we performed a competitive repopulation assay again

using 12-month mice including SB203580 administration sets. As shown in **new version Fig. 5b-f**, GFP-TN13 showed comparable effects with SB203580 administration *in vivo*.

3. The results demonstrate that GFP-TN13 and TAT-TN13 inhibit p38 activity in a very similar manner with SB203580. Are they specific to p38alpha?

What is the advantage to use the newly derived peptide over the traditional p38 inhibitor in term of dosage, efficiency and specificity?

Response:

Reviewer #2 also has raised nearly same comments for TN13 treatment. As shown in our overall results, TN13 is comparable to SB203580 in p38 activity inhibition and rejuvenation of HSCs. TN13 is derived from TXNIP binding sequence of p38 docking site but small molecule inhibitors like SB203580 are competitors of ATP binding site of p38 kinase.

Recent studies have reported the side effects of small molecule inhibitors of p38 from nonspecific targeting (Nat Biotechnol. 2008 Jan;26(1):127-32., J Biol Chem. 2000 Mar 10;275(10):7395-402., British Journal of Pharmacology (2000) 131, 99 – 107). To inhibit signaling pathways in cells, the specific targeting is the most important point. To determine the specificity of TN13, we tried kinase assays for p38 isoforms *in vitro*. As shown in new version **Supplementary Fig. 4i**, SB203580 completely inhibited 2 isoforms, p38 α and p38 β but TN13 showed the specificity on p38 α . It suggests that TN13 may more specific than SB203580 for p38 α inhibition and we may reduce the side effects *in vitro* and *in vivo* by using TN13 as a p38 α inhibitor. And also TN13 is water soluble but SB203580 is DMSO soluble. Drug solubility is also important factor for the application to cells or clinical trials to avoid cytotoxicity in high concentration. We are going to test the side effects of TN13 *in vivo* and *in vitro* further including kinase profiling and toxicity assays.

4. How is the stability of TAT-TN13 peptide in vivo? In order to maintain the rejuvenating effect, what should be the frequency of administration for long term effect?

Response:

1. To determine the stability of TAT-TN13 *in vivo*, we challenged mice with 25 mg/kg FITC labelled-TAT-TN13 by i.v. injection. Then we investigated the fluorescence positive cells from PB using flow cytometry at each time point. As shown in **below figure**, about 24 hours later, most of Fitc-positive cells were disappeared in the PB. So stability of TAT-TN13 is less than 24 hours.

Unpublished Figure 1. Time kinetics of FITC-TAT-TN13 positive cells in PB

2. In our study, we have tried two methods of treating TAT-TN13 peptide. One was a direct injection into mice by i.p. to protect from haematopoietic stress (**new version Supplementary Fig. 7**). The other was treating HSCs with peptide *in vitro* and then injecting peptide-treated HSCs into recipients by i.v. (**new version Fig. 6a-e**). The direct injection of TAT-TN13 may be easier to stimulate HSCs *in vivo* and fully rescued old mice from haematopoietic stress-induced death by daily 4 time injection. We could not observe the obvious adverse effects on mice at least this condition, but long-term treatment of TAT-TN13 may affect other cells and result in side effects by inhibiting p38 activity. To solve this concern, we will optimize the treating schedule in further study. Alternatively, to avoid the side effects from whole body injection, we are going to treat HSCs with peptide *in vitro* after obtaining cells from G-CSF (a HSC mobilization reagent)-treated mice for autologous injection. As shown in **new version Fig 6a-e**, the short-term treatment (16 hours) of TAT-TN13 *in vitro* showed efficient rejuvenation of old HSCs for 4 months in recipients. This approach may be safer than long-term whole body treatment.

5. Recent study suggests that aging is a multifactorial process, so that modulation of aging with a compound to rejuvenate one organ could potentially lead to adverse outcomes in others (Wahlestedt M, 2015). Has the author observed any adverse effect in other organs of the mice after the injection of TXNIP derived peptide? How is the dosage determined?

Response:

1. As mentioned in the reference, whole body treatment of rejuvenating drugs cannot distinguish modulation of aging cells or organ, potentially leading to adverse outcomes in others. Although, we do not have detailed results for adverse effects from TAT-TN13 injection, we could not observe any visible changes or differences in recipients after injecting TAT-TN13 into mice. However long-term treatment of TAT-TN13 may affect other cells and result in side effects by inhibiting p38 activity. As mentioned above (**Reviewer's comment 4**), we will optimize the treating schedule in further study.
2. To determine the dosage of TAT-TN13, we serially administered TAT-TN13 into 12-month-old TXNIP^{-/-} mice and confirmed the inhibition of p38 activity in HSCs using flow cytometry. As shown in **below figure**, the levels of phospho-p38 were decreased by TAT-TN13 in a dose dependent manner. At dosage of 25 mg/kg, the inhibition of p38 activity was maximized and comparable to p38 activity of 2-month-old TXNIP^{+/+} HSCs.

Unpublished Figure 2. Regulation of p38 activity in LT-HSCs by TAT-TN13

Minor concerns

1. Figure 2D seems redundant as it is also shown in part of Figure 2e.

Response:

We have removed Fig. 2d as commented.

2. What is the purpose of introducing GFP-TN13 to rejuvenate HSC besides TAT-TN13? How is the effect compared to TAT-TN13 in parallel?

Response:

We have tried to rejuvenate old HSCs by viral-mediated gene therapy (Lenti-GFP-fused-TN13) as a proof-of-concept to compare with peptide therapy (TAT-TN13). We confirmed the ability of GFP-fused-TN13 on p38 inhibition. As you mentioned in **the Reviewer's comment 2**, we revised the data in **new version Fig. 5b-f**. GFP-fused-TN13 showed the comparable effect with SB203580 administration *in vivo*. So, we could observe the similar effect of GFP-TN13 with TAT-TN13 in HSCs.

3. In Figure 7, in the aged HSC, the statement that 'a loss of TXNIP resulted in p38 activation in HSCs' is not clearly depicted in the illustration, as the number of TXNIP should decrease in aged HSCs that results in an increase of p38 activity.

Response:

Reviewer 1 and 2 also commented on the illustration of TXNIP functions in HSCs. We have illustrated the functions of TXNIP and TN13 in HSCs and revised it in **new version Fig. 7**. As shown in **new version Fig. 7**, TXNIP inhibits p38 activity and activates p53 via direct interaction and these interactions are resulted in the reduction of aging-associated gene expression or the induction of p53 dependent anti-oxidant genes. As a result, TXNIP^{+/+} HSCs can maintain low level of ROS and their homeostasis. However, in TXNIP^{-/-} HSCs, the loss of TXNIP makes them free and ROS-inducing circuit will be turned on easily. At this condition, unregulated ROS may hyper-activate p38 or p53 that induces ageing factors and make HSCs to become old. These ageing phenotypes can be reversed by p38 inhibition via TN13 treatment.

Figure 7. Regulation of ROS in HSCs by TXNIP or TXNIP-derived peptide.

References

Wahlestedt M, Pronk CJ, Bryder D, (2015), Concise review: hematopoietic stem cell aging and the prospects for rejuvenation. *Stem Cells Transl Med.* 4(2):186-94. doi: 10.5966/sctm.2014-0132.

REVIEWERS' COMMENTS:

Reviewer #1 (Remarks to the Author):

No comments.

Reviewer #2 (Remarks to the Author):

The authors have adequately addressed all my concerns. The revised manuscript has been significantly improved to my satisfaction.

Reviewer #3 (Remarks to the Author):

None

Point-by-point response

There are no specific comments from reviewers.